# Social contact patterns and implications for infectious disease transmission – a systematic review and meta-analysis of contact surveys

Andria Mousa[1]*, Peter Winskill[1], Oliver John Watson[2], Oliver Ratmann[2], Mélodie Monod[2], Marco Ajelli[3,4], Aldiouma Diallo[5], Peter J Dodd[6], Carlos G Grijalva[7], Moses Chapa Kiti[8], Anand Krishnan[9], Rakesh Kumar[9], Supriya Kumar[10], Kin O Kwok[11,12,13], Claudio F Lanata[14,15], Olivier Le Polain de Waroux[16], Kathy Leung[17,18], Wiriya Mahikul[19], Alessia Melegaro[20], Carl D Morrow[21,22], Joël Mossong[23], Eleanor FG Neal[24,25], D James Nokes[8,26], Wirichada Pan-ngum[27], Gail E Potter[28,29], Fiona M Russell[24,25], Siddhartha Saha[30], Jonathan D Sugimoto[31,32,33], Wan In Wei[11], Robin R Wood[21], Joseph Wu[17,18], Juanjuan Zhang[34], Patrick Walker[1], Charles Whittaker[1]*

[1]MRC Centre for Global Infectious Disease Analysis, Imperial College London, London, United Kingdom; [2]Department of Mathematics, Imperial College London, London, United Kingdom; [3]Department of Epidemiology and Biostatistics, Indiana University School of Public Health, Bloomington, United States; [4]Laboratory for the Modeling of Biological and Socio-technical Systems, Northeastern University, Boston, United States; [5]VITROME, Institut de Recherche pour le Developpement, Dakar, Senegal; [6]School of Health and Related Research, University of Sheffield, Sheffield, United Kingdom; [7]Division of Pharmacoepidemiology, Department of Health Policy, Vanderbilt University Medical Center, Nashville, United States; [8]KEMRI-Wellcome Trust Research Programme, Kilifi, Kenya; [9]Centre for Community Medicine, All India Institute of Medical Sciences, New Delhi, India; [10]Bill and Melinda Gates Foundation, Seattle, United States; [11]JC School of Public Health and Primary Care, Chinese University of Hong Kong, Hong Kong, China; [12]Stanley Ho Centre for Emerging Infectious Diseases, The Chinese University of Hong Kong, Hong Kong, China; [13]Shenzhen Research Institute of The Chinese University of Hong Kong, Shenzhen, China; [14]Instituto de Investigación Nutricional, Lima, Peru; [15]Department of Medicine, Vanderbilt University, Nashville, United States; [16]London School of Hygiene and Tropical Medicine, London, United Kingdom; [17]WHO Collaborating Centre for Infectious Disease Epidemiology and Control, School of Public Health, LKS Faculty of Medicine, The University of Hong Kong, Hong Kong, China; [18]Laboratory of Data Discovery for Health (D24H), Hong Kong Science Park, Hong Kong, China; [19]Faculty of Medicine and Public Health, HRH Princess Chulabhorn College of Medical Science, Chulabhorn Royal Academy, Bangkok, Thailand; [20]Dondena Centre for Research on Social Dynamics and Public Policy, Department of Social and Political Sciences, Bocconi University, Milano, Italy; [21]Desmond Tutu HIV Centre, Department of Medicine, University of Cape Town, Cape Town, South Africa; [22]Centre for Infectious Disease Epidemiology and Research (CIDER), School of Public Health and Family Medicine, Faculty of Health Sciences, University of Cape Town, Cape Town, South Africa; [23]Health Directorate, Luxembourg, Luxembourg; [24]Infection and

*For correspondence:
a.mousa17@imperial.ac.uk (AM);
charles.whittaker16@imperial.ac.uk (CW)

Immunity, Murdoch Children's Research Institute, Victoria, Australia; [25]Department of Paediatrics, University of Melbourne, Victoria, Australia; [26]School of Life Sciences, University of Warwick, Coventry, United Kingdom; [27]Department of Tropical Hygiene, Faculty of Tropical Medicine, Mahidol University, Bangkok, Thailand; [28]National Institute for Allergies and Infectious Diseases, National Institutes of Health, Rockville, United States; [29]The Emmes Company, Rockville, United States; [30]Influenza Programme, US Centers for Disease Control and Prevention, New Delhi, India; [31]Seattle Epidemiologic Research and Information Center, United States Department of Veterans Affairs, Seattle, United States; [32]Department of Epidemiology, University of Washington, Washington, United States; [33]Fred Hutchinson Cancer Research Center, Seattle, United States; [34]School of Public Health, Fudan University, Key Laboratory of Public Health Safety, Ministry of Education, Shanghai, China

## Abstract

**Background:** Transmission of respiratory pathogens such as SARS-CoV-2 depends on patterns of contact and mixing across populations. Understanding this is crucial to predict pathogen spread and the effectiveness of control efforts. Most analyses of contact patterns to date have focused on high-income settings.

**Methods:** Here, we conduct a systematic review and individual-participant meta-analysis of surveys carried out in low- and middle-income countries and compare patterns of contact in these settings to surveys previously carried out in high-income countries. Using individual-level data from 28,503 participants and 413,069 contacts across 27 surveys, we explored how contact characteristics (number, location, duration, and whether physical) vary across income settings.

**Results:** Contact rates declined with age in high- and upper-middle-income settings, but not in low-income settings, where adults aged 65+ made similar numbers of contacts as younger individuals and mixed with all age groups. Across all settings, increasing household size was a key determinant of contact frequency and characteristics, with low-income settings characterised by the largest, most intergenerational households. A higher proportion of contacts were made at home in low-income settings, and work/school contacts were more frequent in high-income strata. We also observed contrasting effects of gender across income strata on the frequency, duration, and type of contacts individuals made.

**Conclusions:** These differences in contact patterns between settings have material consequences for both spread of respiratory pathogens and the effectiveness of different non-pharmaceutical interventions.

**Funding:** This work is primarily being funded by joint Centre funding from the UK Medical Research Council and DFID (MR/R015600/1).

## Introduction

Previous outbreaks of Ebola (*Mbala-Kingebeni et al., 2019*), influenza (*Khan et al., 2009*), and the ongoing COVID-19 pandemic have highlighted the importance of understanding the transmission dynamics and spread of infectious diseases, which depend fundamentally on the underlying patterns of social contact between individuals. Together, these patterns give rise to complex social networks that influence disease dynamics (*Eubank et al., 2004*; *Ferrari et al., 2006*; *Firth et al., 2020*; *Zhang et al., 2020*), including the capacity for emergent pathogens to become endemic (*Ghani and Aral, 2005*; *Jacquez et al., 1988*), the overdispersion of the offspring distribution underlying the reproduction number (*Delamater et al., 2019*) and the threshold at which herd immunity is reached (*Fontanet and Cauchemez, 2020*; *Mistry et al., 2021*). They can similarly modulate the effectiveness of non-pharmaceutical interventions (NPIs), such as school closures and workplace restrictions, that are typically deployed to control and contain the spread of infectious diseases (*Prem et al., 2020*).

Social contact surveys provide insight into the features of these networks, which is typically achieved through incorporating survey results into mathematical models of infectious disease

**eLife digest** Infectious diseases, particularly those caused by airborne pathogens like SARS-CoV-2, spread by social contact, and understanding how people mix is critical in controlling outbreaks. To explore these patterns, researchers typically carry out large contact surveys. Participants are asked for personal information (such as gender, age and occupation), as well as details of recent social contacts, usually those that happened in the last 24 hours. This information includes, the age and gender of the contact, where the interaction happened, how long it lasted, and whether it involved physical touch.

These kinds of surveys help scientists to predict how infectious diseases might spread. But there is a problem: most of the data come from high-income countries, and there is evidence to suggest that social contact patterns differ between places. Therefore, data from these countries might not be useful for predicting how infections spread in lower-income regions.

Here, Mousa et al. have collected and combined data from 27 contact surveys carried out before the COVID-19 pandemic to see how baseline social interactions vary between high- and lower-income settings. The comparison revealed that, in higher-income countries, the number of daily contacts people made decreased with age. But, in lower-income countries, younger and older individuals made similar numbers of contacts and mixed with all age groups.

In higher-income countries, more contacts happened at work or school, while in low-income settings, more interactions happened at home and people were also more likely to live in larger, inter-generational households. Mousa et al. also found that gender affected how long contacts lasted and whether they involved physical contact, both of which are key risk factors for transmitting airborne pathogens.

These findings can help researchers to predict how infectious diseases might spread in different settings. They can also be used to assess how effective non-medical restrictions, like shielding of the elderly and workplace closures, will be at reducing transmissions in different parts of the world.

transmission frequently used to guide decision making in response to outbreaks (*Chang et al., 2021*; *Davies et al., 2020*). Such inputs are necessary for models to have sufficient realism to evaluate relevant policy questions. However, despite the known importance of contact patterns as determinants of the infectious disease dynamics, our understanding of how they vary globally remains far from complete. Reviews of contact patterns to date have focused on high-income countries (HICs) (*Hoang et al., 2019*). This is despite evidence that social contact patterns differ systematically across settings in ways that have material consequences for the dynamics of infectious disease transmission and the evolution of epidemic trajectories (*Prem et al., 2017*; *Walker et al., 2020*). Previous reviews have also primarily explored the total number of contacts made by individuals (*Hoang et al., 2019*) and/or how these contacts are distributed across different age/sex groups (*Horton et al., 2020*). Whilst these factors are a vital component underpinning disease spread, recent work has also underscored the importance of the characteristics of contacts (such as the location, duration, and extent of physical contact) in determining transmission risk (*Thompson et al., 2021*).

Here, we carry out a systematic review of contact surveys (conducted prior to the emergence of COVID-19) in lower-income, lower-middle and upper-middle-income countries (LICs, LMICs and UMICs, respectively). Alongside previously published data from HICs (*Kwok et al., 2018*; *Kwok et al., 2014*; *Leung et al., 2017*; *Mossong et al., 2008*), we collate individual participant data (IPD) on social contacts from published work spanning 27 surveys from 22 countries and over 28,000 individuals. We use a Bayesian framework to explore drivers and determinants of contact patterns across a wider range of settings and at a more granular scale than has previously been possible. Specifically, we assess the influence of key factors such as age, gender, and household structure on both the total number and characteristics (such as duration, location, and type) of contact made by an individual, and explore how the comparative importance of different factors varies across different settings. We additionally evaluate the extent and degree of assortativity in contact patterns between different groups, and how this varies across settings.

## Materials and methods

### Systematic review

#### Data sources and search strategy

Two databases (Ovid MEDLINE and Embase) were searched on 26 May 2020 to identify studies reporting on contact patterns in LICs, LMICs, and UMICs (*Supplementary file 1*). Collated records underwent title and abstract screening for relevance, before full-text screening using pre-determined criteria. Studies were included if they reported on any type of face-to-face or close contact with humans and were carried out in LICs, LMICs, or UMICs only. No restrictions on collection method (e.g. prospective diary-based surveys or retrospective surveys based on a face-to-face/phone interview or questionnaire) were applied. Studies were excluded if they did not report contacts relevant to air-borne diseases (e.g. sexual contacts), were conducted in HICs, were contact tracing studies of infected cases, or were conference abstracts. All studies were screened independently by two reviewers (AM and CW). Differences were resolved through consensus and discussion. The study protocol can be accessed through PROSPERO (registration number: CRD42020191197). Income group classification (LIC/LMIC, UMIC, or HIC) was based on 2019 World Bank data (fiscal year 2021) (*World Bank Group, 2020*).

#### Data extraction

Individual-level data were obtained from publication supplementary data, as well as online data repositories such as Zenodo, figshare, and OSF. When not publicly available, study authors were contacted to request data. Extracted data included the participant's age, gender, employment, student status, household size, and total number of contacts, as well as the day of the week for which contacts were reported. Some studies reported information at the level of individual contacts and included the age, gender, location, and duration of the contact, as well whether it involved physical contact. Individual-level data from HICs, not systematically identified, were used for comparison, and included three studies from Hong Kong (*Kwok et al., 2018*; *Kwok et al., 2014*; *Leung et al., 2017*) and the eight European countries from the POLYMOD study (*Mossong et al., 2008*). Data were collated, cleaned, and standardised using Stata version 14. Country-specific average household size was obtained from the United Nations Database on Household Size and Composition (*United Nations Department of Economic and Social Affairs Population Division, 2019*). Gross domestic product based on purchasing power parity (GDP PPP) was obtained from the World Data Bank database (*World Bank International Comparison Programme, 2021*). Findings are reported in accordance with the Preferred Reporting Items for Systematic Reviews and Meta-Analyses (PRISMA) checklist of items specific to IPD meta-analyses (*Supplementary file 2*). Risk of bias was assessed using the AXIS critical appraisal tool used to evaluate quality of cross-sectional studies (*Downes et al., 2016*), modified to this study's objectives (*Supplementary file 3*). Each item was attributed a zero or a one, and a quality score was assigned to each study, ranging from 0% ('poor' quality) to 100% ('good' quality). The individual-level data across all studies and analysis code are available at https://github.com/mrc-ide/contact_patterns (*Whittaker, 2021*; copy archived at swh:1:rev:0b732099d66b2788ae-6da5cf0e8185b25de70868; see *Supplementary file 4* for data assumptions and *Supplementary file 5* for data dictionary).

### Statistical analysis

The mean, median, and interquartile range of total daily unique contacts were calculated for subgroups including country income status, individual study, survey methodology (diary-based or questionnaire/interview-based), survey day (weekday/weekend), and respondent characteristics such as age, sex, employment/student status, and household size. Detailed description of data assumptions for each study can be found in *Supplementary file 4*.

A negative binomial regression model was used to explore the association between the total number of daily contacts and the participant's age, sex, employment/student status, and household size, as well as methodology and survey day. Incidence rate ratios from these regressions are referred to as 'contact rate ratios' (CRRs). A sensitivity analysis was carried out that excluded additional contacts (such as additional work contacts, group contacts, and number missed out, which were recorded separately and in less detail by participants compared to their other contacts [*Ajelli and Litvinova, 2017*; *Kumar et al., 2018*; *Leung et al., 2017*; *Zhang et al., 2020*]). Logistic regressions

were used to explore determinants of contact duration (<1 hr/1 hr+) and type (physical/non-physical), using the same explanatory variables as in the total contacts analyses. There were differences in the contact duration categories defined by studies, and the threshold of 1 hr for longer durations was used to maximise sample size, by allowing inclusion of all available data. An additional sensitivity analysis, weighing all studies equally within an income stratum, explored the impact of study size on the estimated CRRs and ORs for all main outcomes (total contacts, duration, and whether physical). The proportion of contacts made at each location (home, school, work, and other) was explored descriptively and contacts made with the same individual in separate locations/instances were considered as separate contacts.

All analyses were done in a Bayesian framework using the probabilistic programming language Stan, using uninformative priors in all analyses and implemented in R via the package *brm*s (**Bürkner, 2018**, **Bürkner, 2017**). All analyses were stratified by three income strata (LICs and LMICs were combined to preserve statistical power) and included random effects by study, to account for heterogeneity between studies. The only exceptions to this were any models adjusting for methodology which did not vary by study. The effect of each factor was explored in an age- and gender-adjusted model. All models exploring the effect of student status or employment status were restricted to children aged between 5 and 18 years and adults over 18, respectively. In the remaining models including all ages, age was adjusted as a categorical variable (< 15, 15–65, and over 65 years). CRRs, odds ratios (ORs), and their associated 95% credible intervals are presented for all regression models. Here, we report estimates adjusted for age and gender (referred to as adjCRR or adjOR). Studies which collated contact-level data were used to assess assortativity of mixing by age and gender for different country-income strata by calculating the proportions of contacts made by participants that are male or female and those that belong to three broad age groups (children, adults, and older adults).

## Results
### Systematic review and IPD meta-analysis
A total of 3409 titles and abstracts were retrieved from the databases, and 313 full-text articles were screened for eligibility (*Appendix 1—figure 1*). This search identified 19 studies with suitable contact data from LIC, LMIC, and UMIC settings – individual-level data were obtained from 16 of these studies, including one study from an LIC, six studies from an LMIC, and nine studies from an UMIC. These were analysed alongside four HIC studies from Hong Kong and Europe. The majority of the studies collected data representative of the general population, through random sampling and included a combination of both rural and urban sites (see Appendix 1 for further details). Although most studies included respondents of all ages, one study restricted their participants to ages over 18 years (*Dodd et al., 2016*), one to ages over 15 years (*Mahikul et al., 2020*), one to ages over 6 months (*Huang et al., 2020*), one study only collected contact data on infants under 6 months (*Oguz et al., 2018*), and another on contacts of children under 6 years and their caregivers (*Neal et al., 2020*). The distribution of participant age groups in each study was also dependent on the sampling method. For instance, two studies focused on school and university students and their contacts, thereby oversampling older children and young adults (*Ajelli and Litvinova, 2017*; *Stein et al., 2014*). Details of the identified studies and a full description of the systematic review findings can be found in Appendix 1 and *Supplementary file 6*.

In total, this meta-analysis yielded 28,503 participants reporting on 413,069 contacts. All studies contained information on main demographic variables such as age and gender. Availability of other variables analysed here for each study are listed in *Supplementary file 7*. All studies reported the number of contacts made in the past 24 hr of (or day preceding) the survey. The definitions of contacts were broadly similar across studies (*Supplementary file 6*). Specifically, contacts were defined as skin-to-skin (physical) contact or a two-way conversation in the physical presence of another person. All studies scored above 65% of the items on the AXIS risk of bias tool, suggesting good or fair quality (*Supplementary file 3*). Among all participants 47.5% were male, 30.1% were aged under 15 years and 7.2% were aged over 65 years. The majority (83.4%) of participants were asked to report the number of contacts they made on a weekday. A large proportion (34.1%) of respondents lived in large households of six or more people but this was largely dependent on income setting (LIC/LMIC =

63.2%, UMIC = 35.9%, HIC = 4.9%). Among school-aged children (5–18 years), 88.1% were students, and 59.1% of adults aged over 18 were employed.

## Total number of contacts and contact location

The median number of contacts made per day across all the studies was 9 (IQR = 5–17), and was similar across income strata (LIC/LMIC = 10[5–17], UMIC = 8[5–16], HIC = 9[5–17]; *Table 1*). There was a large variation in contact rates across different studies, with the median number of daily contacts ranging from 4 in a Zambian setting (*Dodd et al., 2016*) to 24 in an online Thai survey (*Stein et al., 2014*). When stratifying by study methodology, median daily contacts was higher in diary-based surveys compared to interview-/questionnaire-based surveys, which was true across all income strata (*Table 1*, *Appendix 2—figure 1*).

Overall, children aged 5–15 had the highest number of daily contacts (*Figure 1A–C*), although there was substantial variation between studies and across income strata in how the number of daily contacts varied with age (*Figure 1A–C*). Across UMICs and HICs, the number of daily contacts made by participants decreased with age, with this decrease most notable in the oldest age groups (adjCRR for 65+ vs. <15 years [95%CrI]: UMIC = 0.67[0.63–0.71] and HIC = 0.57[0.54–0.60]). By contrast, there was no evidence of contact rates declining in the oldest age groups in LICs/LMICs (adjCRR for 65+ vs. < 15 years [95%CrI] = 0.94[0.89–1.00]). We observed contrasting effects of gender on the number of daily contacts, with men making more daily contacts compared to women in LICs/LMICs after accounting for age (adjCRR = 1.17, 95%CrI:1.15–1.20; *Figure 1D*), but no effect of gender on total daily contacts for other income strata (CRR[95%CrI]: UMIC = 1.01[0.98–1.04], HIC = 0.99[0.97–1.02]). There were also differences in the number of daily contacts made according to the methodology used and whether the survey was carried out on a weekday or over the weekend – in both instances, contrasting effects of these factors on the number of daily contacts according to income strata were observed (*Figure 1D,F*).

We also examined the influence of factors that might influence both the total number and location (home, work, school, and other) of the contacts individuals make. Across all income strata, students (defined as those currently in education, attending school, and aged between 5 and 18 years) made more daily contacts than non-students aged between 5 and 18 (adjCRR [95%CrI]: LIC/LMIC = 1.26[1.16–1.37], UMIC = 1.18[1.03–1.35] and HIC = 1.54[1.42–1.66]; *Figure 1D–F*). Similarly, we observed strong and significant effects of employment in all income strata, with adults who were employed having a higher number of total daily contacts compared to those not in employment (adjCRR [95%CrI]: LIC/LMIC = 1.17[1.12–1.23], UMIC = 1.07[1.03–1.13], HIC = 1.60[1.54–1.65]; *Figure 1D–F*). The number of daily contacts made at home was proportional to the participant's household size (*Appendix 2—figure 2*). Total daily contacts increased with household size (*Figure 2A*, *Appendix 2—figure 1*) across all income-strata; individuals living in large households (6+ members) had 1.47 (95%CrI:1.32–1.64) (LIC/LMICs), 2.58 (95%CrI:2.37–2.80) (UMICs), and 1.51 (95%CrI:1.40–1.63) (HICs) times more daily contacts than those living alone, after accounting for age and gender (*Figure 1E–F*). Sensitivity analyses excluding additional contacts (as defined in Materials and methods) showed little difference in effect sizes for total daily contacts, and were strongly correlated with the effect sizes shown in *Figure 1D–F* (*Appendix 2—figure 3*).

Motivated by this suggestion of strong, location-related (school, work, and household) effects on total daily contact rates, we further explored the locations in which contacts were made. Contact location was known for 314,235 contacts, 42.7% of which occurred at home (13.1% at work, 12.5% at school, and 31.7% in other locations). Across income strata, there was significant variation in the proportion of contacts made at home – being highest in LICs/LMICs (68.3%) and lowest in HICs (37.0%) (*Figure 2B*). Age differences were also observed in the number of contacts made at home, particularly for LICs/LMICs (*Figure 2C,D*). Relatedly, a higher proportion of contacts occurred at work and school (14.6% and 11.3%) in HICs compared to LICs/LMICs (3.9% and 5.2%, respectively; *Appendix 2—figure 4*). Strong, gender-specific patterns of contact location were also observed. Across all income strata males made a higher proportion of their contacts at work compared to females, although this difference was largest for LICs/LMICs (*Appendix 2—figure 4* and *Appendix 2—figure 5*). Further, we found significant variation between income strata in median household size (seven in LICs/LMICs, five in UMICs, and three in HICs). This trend of decreasing household size with increasing country income was consistent with global data (*Figure 2E*). The larger households observed for LIC/LMIC settings

**Table 1.** Summary table of total daily contacts.

The total number of observations, as well as the mean, median, and interquartile range (p25 and p75) of total daily contacts shown by participant and study characteristics.

| | | | N | Mean | p25 | Median | p75 |
|---|---|---|---|---|---|---|---|
| Overall | | | 28,503 | 14.5 | 5 | 9 | 17 |
| Gender | | Male | 13,218 | 15.3 | 5 | 9 | 18 |
| | | Female | 14,598 | 13.7 | 5 | 9 | 16 |
| Age | | <15 | 8,561 | 14.6 | 6 | 10 | 19 |
| | | 15–65 | 17,841 | 14.9 | 5 | 9 | 17 |
| | | >65 | 2,047 | 10.4 | 3 | 6 | 12 |
| Income status | | LIC/LMIC | 9,906 | 15.4 | 5 | 10 | 17 |
| | | UMIC | 8,330 | 14.4 | 5 | 8 | 16 |
| | | HIC | 10,267 | 13.7 | 5 | 9 | 17 |
| Survey Methodology | | Diary | 12,226 | 13.9 | 6 | 10 | 18 |
| | | Interview/survey | 16,227 | 15.0 | 4 | 8 | 16 |
| Day type | | Weekend | 4,308 | 14.7 | 5 | 9 | 16 |
| | | Weekday | 21,579 | 14.1 | 5 | 9 | 17 |
| Employment | | Yes | 8,879 | 15.4 | 5 | 9 | 17 |
| (*in those aged > 18*) | | No | 6,158 | 9.8 | 4 | 7 | 12 |
| Student | | Yes | 4,438 | 18.4 | 8 | 14 | 24 |
| (*in those aged 5–18*) | | No | 600 | 10.4 | 5 | 8 | 14 |
| Household size | | 1 | 1,479 | 10.4 | 3 | 6 | 12 |
| | | 2 | 3,220 | 11.8 | 4 | 7 | 14 |
| | | 3 | 4,130 | 12.0 | 4 | 7 | 14 |
| | | 4 | 5,240 | 13.4 | 5 | 8 | 17 |
| | | 5 | 3,109 | 12.5 | 4 | 8 | 14 |
| | | 6+ | 8,873 | 17.7 | 7 | 11 | 20 |
| Study | Belgium | Mossong | 750 | 11.8 | 5 | 9 | 15 |
| | China | Read | 1,821 | 18.6 | 7 | 13 | 22 |
| | China | Zhang | 965 | 18.8 | 4 | 10 | 30 |
| | Fiji | Neal | 2,019 | 6.4 | 4 | 6 | 8 |

*Table 1 continued on next page*

*Table 1 continued*

| | | N | Mean | p25 | Median | p75 |
|---|---|---|---|---|---|---|
| Finland | Mossong | 1,006 | 11.1 | 5 | 9 | 15 |
| Germany | Mossong | 1,341 | 7.9 | 4 | 6 | 10 |
| Hong Kong | *Kwok et al., 2014* | 762 | 18.3 | 5 | 9 | 18 |
| Hong Kong | *Kwok et al., 2018* | 1,066 | 11.9 | 3 | 7 | 13 |
| Hong Kong | Leung | 1,149 | 14.4 | 3 | 7 | 15 |
| India | Kumar | 2,943 | 27.0 | 12 | 17 | 26 |
| Italy | Mossong | 849 | 19.8 | 10 | 17 | 27 |
| Kenya | Kiti | 568 | 17.7 | 10 | 15 | 23 |
| Luxembourg | Mossong | 1,051 | 17.5 | 8 | 14 | 24 |
| The Netherlands | Mossong | 269 | 13.9 | 6 | 11 | 19 |
| Peru | Grijalva | 588 | 15.3 | 8 | 12 | 20 |
| Poland | Mossong | 1,012 | 16.3 | 7 | 13 | 22.5 |
| Russia | Ajelli | 502 | 18.0 | 6 | 11 | 19 |
| South Africa | Dodd | 1,276 | 5.2 | 4 | 5 | 7 |
| South Africa | Wood | 571 | 15.6 | 9 | 14 | 20 |
| Senegal | Potter | 1,417 | 19.7 | 10 | 15 | 25 |
| Thailand | Mahikul | 369 | 22.6 | 13 | 20 | 31 |
| Thailand | Stein | 219 | 58.5 | 15 | 24 | 55 |
| Uganda | Le Polain de Waroux | 568 | 7.0 | 5 | 7 | 9 |
| United Kingdom | Mossong | 1,012 | 11.7 | 6 | 10 | 16 |
| Vietnam | Horby | 865 | 7.7 | 5 | 7 | 9 |
| Zambia | Dodd | 2,300 | 4.8 | 3 | 4 | 6 |
| Zimbabwe | Melegaro | 1,245 | 10.7 | 6 | 9 | 14 |

were also more likely to be intergenerational – in LICs/LMICs, 59.4% of participants aged over 65 lived in households of at least six members compared to 17.5% in UMICs and only 2.2% in HICs.

## Type and duration of contact

Data on the type of contacts (physical and non-physical) were recorded for 20,910 participants. The mean percentage of physical contacts across participants was 56.0% and was the highest for LICs/ LMICs (64.5%). At the study level, the highest mean percentage of physical contacts was observed for a survey of young children and their caregivers conducted in Fiji (*Neal et al., 2020*) (84.0%) and the lowest in a Hong Kong contact survey (*Leung et al., 2017*) (18.9%). Physical contact was significantly less common among adults compared to children under 15 years in all settings (ORs ranged between 0.22 and 0.48) (*Figure 3A–F*). Despite the proportion of physical contacts generally decreasing with age, there was a higher proportion observed for adults aged 80 or over (*Figure 3A–C*). Contacts made by male participants were more likely to be physical compared to female participants in UMICs (adjOR = 1.13, 95%CrI = 1.10–1.16) and HICs (adjOR = 1.09, 95%CrI = 1.07–1.12), but in LICs/LMICs men had a lower proportion of physical contacts than women (adjOR = 0.81, 95%CrI = 0.79–0.83; *Figure 3D–F*). Most physical contacts made by women in LICs were made at home (73.5%), whilst for HICs this was just 41.4% – similar differences across income strata were observed for men, although the proportions were always lower than observed for women (62.4% for LIC/LMICs and 36.4% for HICs). Increasing household size was generally associated with a higher proportion of contacts being

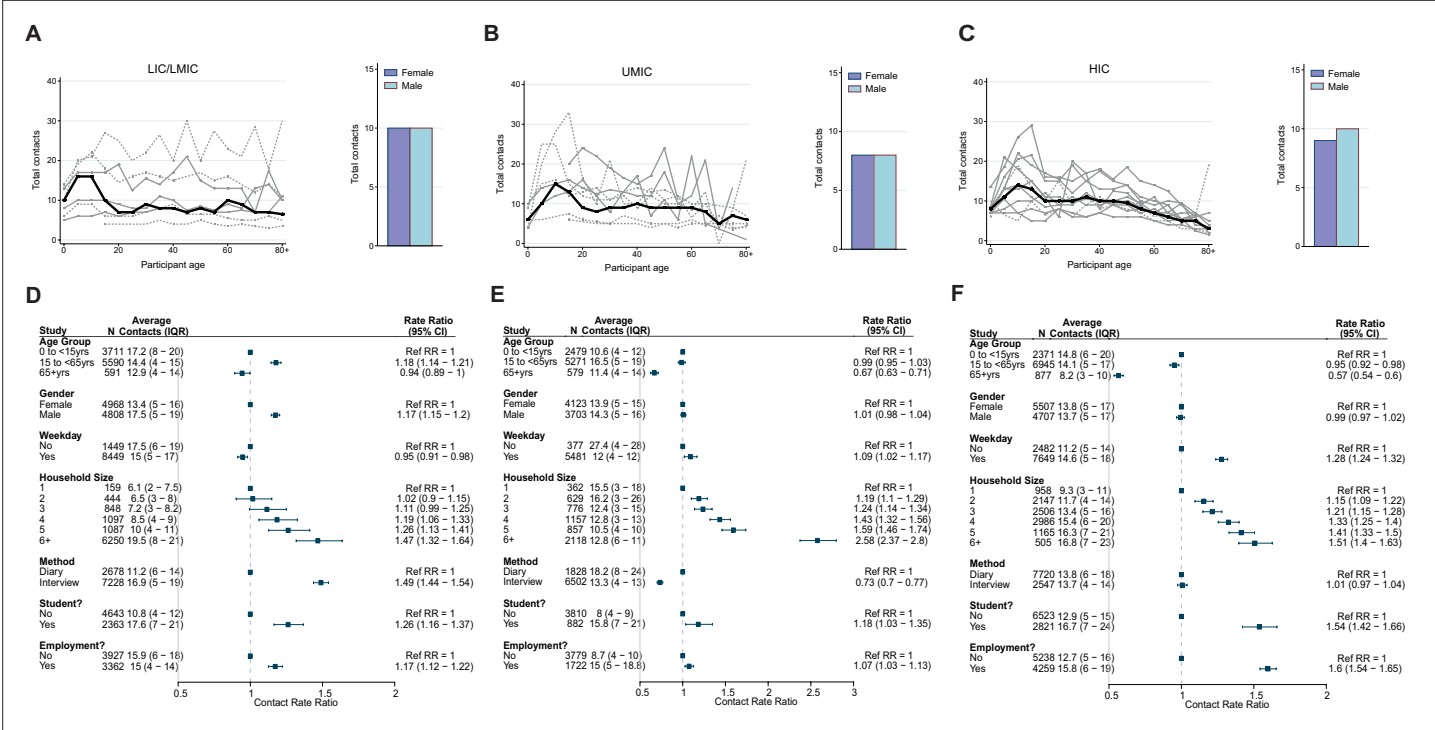

**Figure 1.** Total number of contacts. Sample median total number of contacts shown by gender (right) and 5-year age groups up to ages 80+ shown for (**A**) lower-income countries (LICs)/lower-middle countries (LMICs), (**B**) upper-middle-income countries (UMICs), and (**C**) high-income countries (HICs). Grey lines denote individual studies, and the solid black line is the median across all studies of within that income group. Studies with a diary-based methodology are represented by a solid grey line and those with a questionnaire or interview design are shown as a dashed line. For UMICs, one study outlier with extremely high number of contacts is excluded (online Thai survey with a 'snowball' design by **Stein et al., 2014**). Contact rate ratios and associated 95% credible intervals from a negative binomial model with random study effects are shown in (**D**) (LICs/LMICs), (**E**) (UMICs), and (**F**) (HICs). All models were adjusted for age and gender and were ran separately for each key variable (weekday/weekend, household size, survey methodology, student/employment status).

physical (for households of 6+ members compared to one member: adjCRR[95%CrI]: LIC/LMIC = 1.73[1.48–2.02], UMIC = 1.30[1.12–1.52], HIC = 1.57[1.48–1.67]; **Figure 3D–F**). Employment was associated with having a significantly lower proportion of physical contacts in LICs/LMICs (adjOR = 0.83, 95%CrI:0.79–0.87) and HICs (adjOR = 0.71, 95%CrI:0.69–0.73), but not in UMICs (adjOR = 1.11, 95%CrI:1.03–1.19). The proportion of physical contacts among all contacts was the highest for households (70.4%), followed by schools (58.5%), community (55.7%), and work (33.6%) (**Appendix 2— figure 6**).

Data on the duration of contact (<1 or ≥1 hr) were available for 22,822 participants. The percentage of contacts lasting at least 1 hr was 63.2% and was highest for UMICs (76.0%) and lowest for LICs/ LMICs (53.1%). Across both UMICs and HICs, duration of contacts was lower in individuals aged over 15 years compared to those aged 0–15, with the extent of this disparity most stark for HICs (for ages 65+ compared to <15 years: adjCRR [95%CrI]: LIC/LMIC = 0.61[0.57–0.64], UMIC = 0.61[0.58– 0.65], HIC = 0.35[0.33–0.37]; **Figure 4A–F**). We observed contrasting effects of gender across income strata: males made longer-lasting contacts than females in UMICs (adjOR = 1.11, 95%CrI = 1.08–1.14; **Figure 4D–F**), but not in LIC/LMICs (adjOR = 0.92, 95%CrI = 0.90–0.95) or HICs (adjOR = 0.98, 95%CrI = 0.97–1.00). Participants reported shorter contacts on weekends compared to weekdays in LICs/LMICs (adjOR = 0.91, 95%CrI: 0.88–0.95), and HICs (adjOR = 0.95, 95%CrI: 0.92–0.97), but not in UMICs (adjOR = 1.12, 95%CrI = 1.03–1.21). Contacts lasting over an hour as a proportion of all contacts was highest for households (72.7%), followed by schools (67.9%), community (47.0%), and work (44.0%). However, it was only in HICs that there was a significant effect of being a student (adjOR = 1.18, 95%CrI: 1.09–1.27; **Figure 4D–F**) on the proportion of contacts lasting ≥1 hr. For all income strata, the proportion of contacts >1 hr increased with increasing household size (**Figure 4D–F**). The sensitivity analysis weighing all studies equally within an income group yielded similar results to those

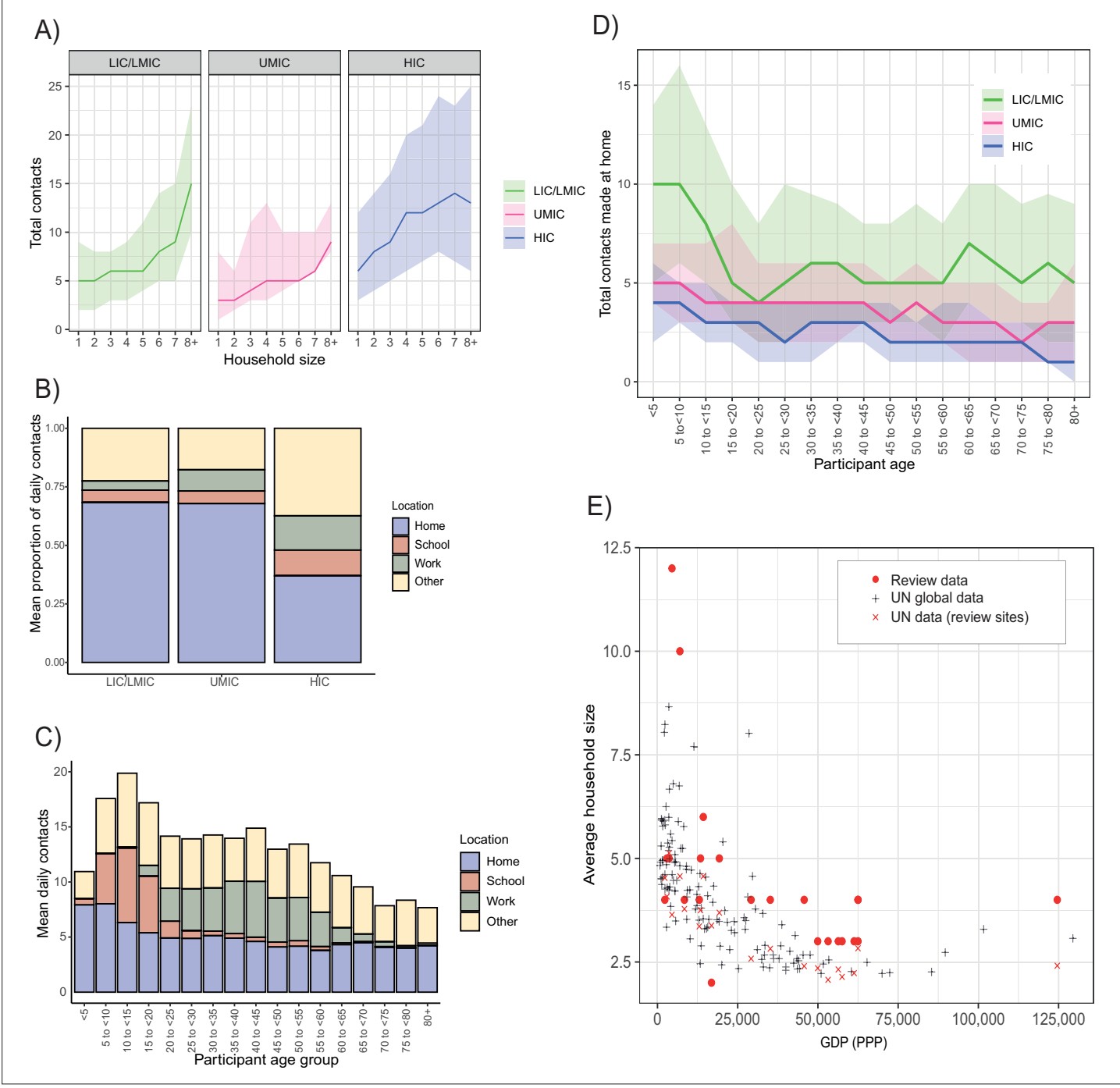

**Figure 2.** Contact location and household size. (**A**) Sample median number of contacts by household size in review data, stratified by income strata. Shaded area denotes the interquartile range. (**B**) Sample mean % of contacts made at each location (home, school, work, other) by income group. (**C**) Total daily contacts (sample mean number) made at each location by 5-year age group. (**D**) Sample median number of contacts made at home by 5-year age groups and income strata. Shaded area denotes the interquartile range. (**E**) Average household size and GDP; red circles represent median household size in single studies from the review. GDP information was obtained from the World Bank Group and global household size data from the Department of Economic and Social Affairs, Population Division, United Nations.

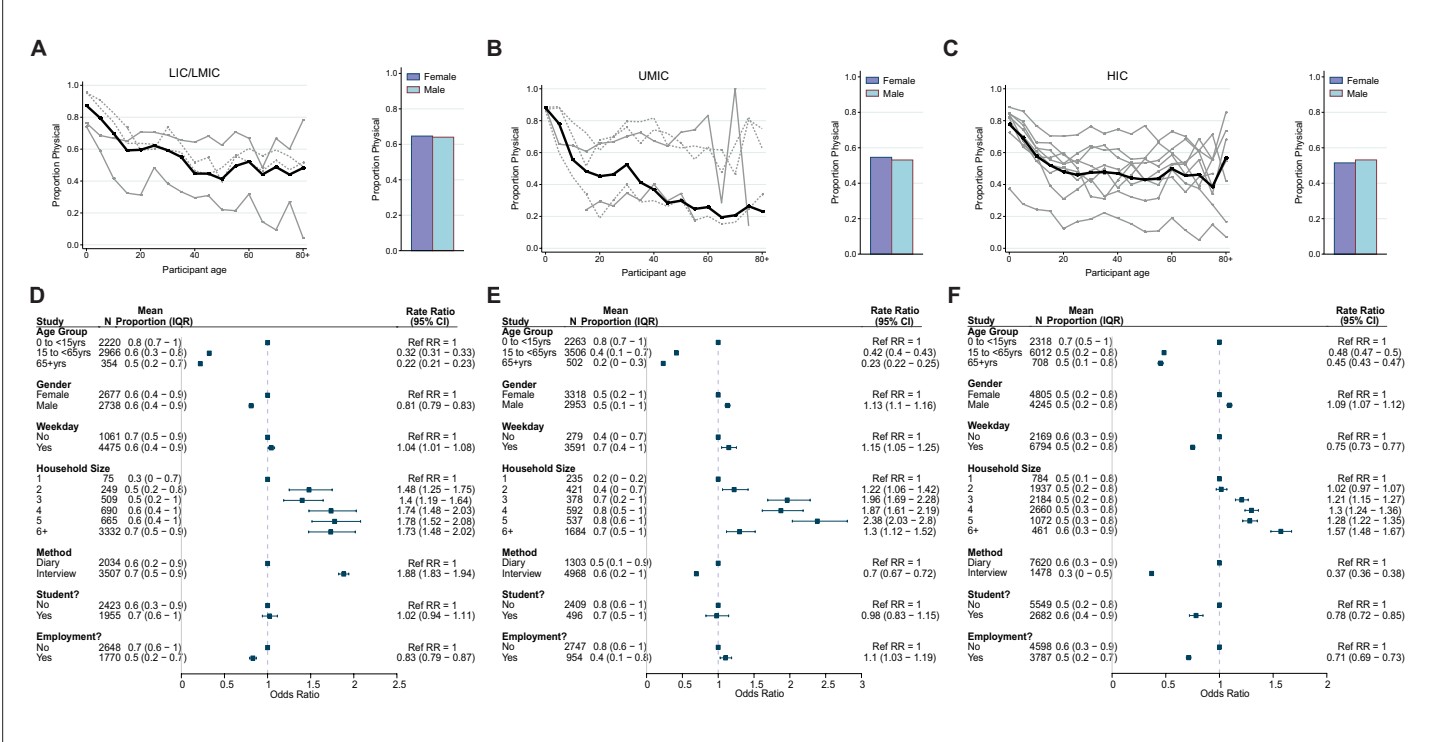

**Figure 3.** Physical contacts. Mean proportion of contacts that are physical shown by gender (right) and 5-year age groups up to ages 80+ shown for (**A**) lower-income countries (LICs)/lower-middle countries (LMICs), (**B**) upper-middle-income countries (UMICs), and (**C**) high-income countries (HICs). Grey lines denote individual studies, and the solid black line is the mean across all studies of within that income group. Studies with a diary-based methodology are represented by a solid grey line and those with a questionnaire or interview design are shown as a dashed line. Odds ratios and associated 95% credible intervals from a logistic regression model with random study effects are shown in D (LICs/LMICs), E (UMICs), and F (HICs). All models were adjusted for age and gender and were ran separately for each key variable (weekday/weekend, household size, survey methodology, student/employment status).

from the main analysis (range of Pearson's correlation coefficients between main analysis and sensitivity analysis effect sizes: 0.92–1.00; *Appendix 2—figure 7* and *Appendix 2—table 1*), and any differences are discussed in Appendix 2.

## Assortativity by age and gender

Twelve studies collected information on the gender of the contact and eight studies contained information on age allowing assignment of contacts to one of the three age groups described in Materials and methods (Appendix 2). We found evidence to suggest that contacts were assortative by gender for all income strata, as participants were more likely to mix with their own gender (*Appendix 2—table 2* and *Appendix 2—table 3*). Mixing was also assortative by age, with participants more likely to contact individuals who belonged to the same age group this degree of age assortativity was lowest for LICs/LMICs, where only 29% of contacts made by adults were with individuals of the same age group. By contrast, in HICs we observed a higher degree of assortative mixing, with most contacts (51.4%) made by older adults occurring with individuals belonging to the same age group.

## Discussion

Understanding patterns of contact across populations is vital to predicting the dynamics and spread of infectious diseases, as well understanding the control interventions likely to have the greatest impact. Here, using a systematic review and individual-participant data meta-analysis of contact surveys, we summarise research exploring these patterns across a range of populations spanning 28,503 individuals and 22 countries. Our findings highlight substantial differences in contact patterns between income settings. These differences are driven by setting-specific sociodemographic factors such as

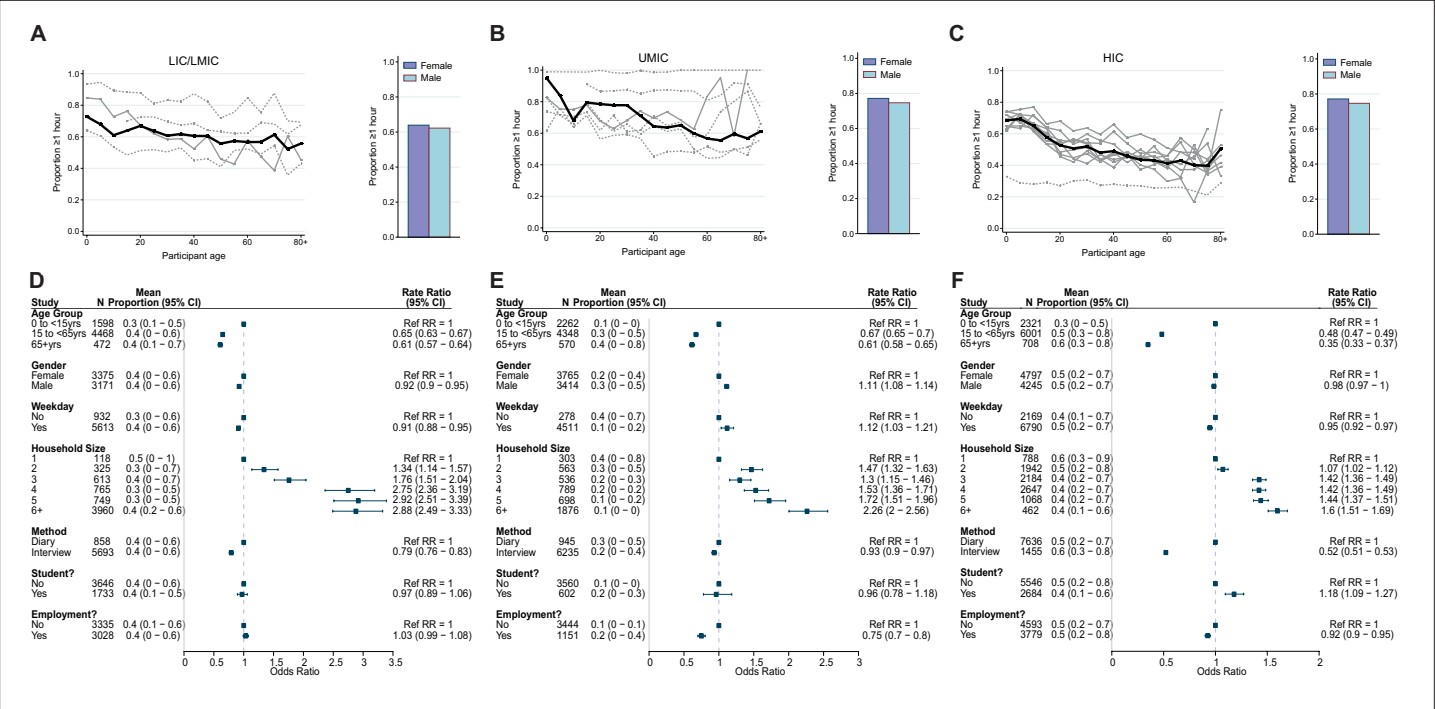

**Figure 4.** Contact duration. Mean proportion of contacts that last at least an hour shown by gender (right) and 5-year age groups up to ages 80+ shown for (A) lower-income countries (LICs)/lower-middle countries (LMICs), (B) upper-middle-income countries (UMICs), and (C) high-income countries (HICs). Grey lines denote individual studies and the solid black line is the mean across all studies of within that income group. Studies with a diary-based methodology are represented by a solid grey line and those with a questionnaire or interview design are shown as a dashed line. Odds ratios and associated 95% credible intervals from a logistic regression model with random study effects are shown in (D) (LICs/LMICs), (E) (UMICs), and (F) (HICs). All models were adjusted for age and gender and were ran separately for each key variable (weekday/weekend, household size, survey methodology, student/employment status).

age, gender, household structure, and patterns of employment, which all have material consequences for transmission and spread of respiratory pathogens.

Across the collated studies, the total number of contacts was highest for school-aged children. This is consistent with previous results from HICs (*Béraud et al., 2015*; *Fu et al., 2012*; *Hoang et al., 2019*; *Ibuka et al., 2016*; *Lapidus et al., 2013*) and shown here to be generally true for LICs/LMICs and UMICs also. Interestingly however, we observed differences in patterns of contact in adults across income strata. Whilst contact rates in HICs declined in older adults, this was not observed in LICs/LMICs, where contact rates did not differ in the oldest age group compared to younger ages. This is consistent with variation in household structure and size across settings, with nearly two-thirds of participants aged 65+ in included LIC/LMIC surveys living in large, likely intergenerational, households (6+ members), compared to only 2% in HICs. HICs were also characterised by more assortative mixing between age groups, with older adults in LICs/LMICs more likely to mix with individuals of younger ages, again consistent with the observed differences between household structures across the two settings. These results have important consequences for the viability and efficacy of protective policies centred around shielding of elderly individuals (i.e. those most at risk from COVID-19 or influenza). In these settings other strategies may be required to effectively shield vulnerable populations, as has been previously suggested (*Dahab et al., 2020*). Our results support the idea of households as a key site for transmission of respiratory pathogens (*Thompson et al., 2021*), with the majority of contacts made at home. Our analysis highlights that the number of contacts made at home is mainly driven by household size. However, the relative importance of households compared to other locations is likely to vary across settings. We observed significant differences across income settings in the distribution of contacts made at home, work, and school. The proportion of contacts made at home was highest for LIC/LMICs, where larger average household sizes were associated with more contacts, more physical contacts, and longer lasting contacts. By contrast, participants in HICs tended to report

more contacts occurring at work and school. The lower number of contacts at work in LIC/LMIC may be explained by the types of employment (e.g. agriculture in rural surveys) and a selection bias (women at home/homemakers more likely to be surveyed in questionnaire-based surveys). Our analyses similarly highlighted significant variation in the duration and nature of contacts across settings. Contacts made by female participants in LICs/LMICs were more likely to be physical compared to men, whilst the opposite effect was observed for HICs and UMICs, potentially reflecting context-specific gender roles. In all settings, we observed a general decline of physical contacts with age, except in the very old (*Mossong et al., 2008*), potentially reflecting higher levels of dependency and the need for physical care.

Altogether, these results suggest differences between settings in the comparative importance of different locations (such as the household or the workplace) to transmission of SARS-CoV-2, a finding which would likely modulate the impact of different NPIs (such as workplace or school closures, stay at home orders, etc.). Moreover, it suggests that previous estimates of NPI effectiveness primarily derived from European data and settings (*Brauner et al., 2021*) may be of limited generalisability to non-European settings characterised by different structures and patterns of social contact. However, beyond highlighting heterogeneity in where and how transmission is likely to occur, it remains challenging to disentangle exactly how these differences in contact patterns would shape patterns of transmission. Whilst the collated data provide a cross-sectional snapshot into the networks of social contact underpinning transmission, they remain insufficient to completely resolve this network or its temporal dynamics. Our results therefore do not consider key features relevant to population-level spread and transmission (such as overall network structure or the extent of repeated contacts, which would be most likely to occur with household members) which previous work has demonstrated can have a significant impact on infectious disease dynamics, both in general terms (*Bansal et al., 2010*; *Keeling and Eames, 2005*) and with COVID-19 (*Rader et al., 2020*). It is in this context that recent results generating complete social networks (including both the frequency and identity of an individual's contacts) from high-resolution GPS data represent promising developments in understanding social contact networks and how they shape transmission (*Firth et al., 2020*).

There are important caveats to these findings. Data constraints limited the numbers of factors we were able to explore – for example, despite evidence (*Kiti et al., 2014*) suggesting that contact patterns differ across rural and urban settings, only three studies (*Kiti et al., 2014*; *le Polain de Waroux et al., 2018b*; *Neal et al., 2020*) contained information from both rural and urban sites, allowing classification. Similarly, we were unable to examine the impact of socioeconomic factors such as household wealth, despite experiences with COVID-19 having highlighted strong socioeconomic disparities in both transmission and burden of disease (*De Negri et al., 2021*; *Routledge et al., 2021*; *Ward et al., 2021*; *Winskill et al., 2020*) and previous work suggesting that poorer individuals are less likely to be employed in occupations amenable to remote working (*Loayza, 2020*). A lack of suitably detailed information in the studies conducted precludes analysis of these factors but highlights the importance of incorporating economic questions into future contact surveys, such as household wealth and house square footage. Other factors also not controlled for here, but that may similarly shape contact patterns include school holidays or seasonal variations in population movement and composition that we are unable to capture given the cross-sectional nature of these studies.

Another important limitation to these results is that we are only able to consider a limited set of contact characteristics (the location and duration of the contact and whether it was physical). Previous work has highlighted the importance of these factors in determining the risk of respiratory pathogen transmission (*Chang et al., 2021*; *Dunne et al., 2018*; *le Polain de Waroux et al., 2018a*; *Neal et al., 2020*; *Thompson et al., 2021*), but only a limited number of studies reported whether a contact was 'close' or 'casual' (*Kwok et al., 2018*; *Kwok et al., 2014*; *le Polain de Waroux et al., 2018b*) and whether the contact was made indoors or outdoors (*Wood et al., 2012*); both factors likely to influence transmission risk (*Bulfone et al., 2021*; *Chu et al., 2020*). More generally, the relevance and comparative importance of different contacts to transmission likely varies according to the specific pathogen and its predominant transmission modality (e.g. aerosol, droplet, fomite, etc.). It is therefore important to note that these results do not provide a direct indication of explicit transmission risk, but rather an indicator of factors likely to be relevant to transmission.

Relatedly, it is also important to note that the studies collated here were conducted over a wide time-period (2005–2018). In conjunction with the cross-sectional nature of the included studies, this

precludes us from being able to examine for potential time-related trends in contact patterns. Additionally, the collated surveys were all carried out prior to the onset of the SARS-CoV-2 pandemic. Previous work has documented significant alterations to patterns of social contact in response to individual-level behaviour changes or government implemented NPIs aimed at controlling SARS-CoV-2 spread, and that these changes are dynamic and time-varying (*Gimma et al., 2021*; *McCreesh et al., 2021*). A detailed understanding of the impact of changing contact patterns on disease spread necessarily requires both an understanding of baseline contact patterns (as detailed in the studies collated here) and what changes have occurred as a result of control measures – however this latter data remains sparse and is available for only a limited number of settings (*Jarvis et al., 2021*; *Jarvis et al., 2020*; *Liu et al., 2021*). Description of contact location was also coarse and precluded more granular analyses of specific settings, such as markets, which have previously been shown to be important locations for transmission in rural areas (*Grijalva et al., 2015*).

Heterogeneity between studies was larger for LICs/LMICs and UMICs, which we partly accounted for, through fitting random study effects. These study differences may be attributed to the way individual contact surveys were conducted, making comparisons of contact patterns among surveys more difficult (e.g. prospective/retrospective diary surveys, online/paper questionnaires, face-to-face/phone interviews, and different contact definitions). For instance, there is evidence suggesting that prospective reporting, which is less affected by recall bias, can often lead to a higher number of contacts being reported (*Mikolajczyk and Kretzschmar, 2008*) and a lower probability of casual or short-lasting contacts being missed. The relatively high contact rates observed in HICs may be explained by the fact that all but two HIC surveys used diary methods. Our study highlights that a unified definition of 'contact' and standard practice in data collection could help increase the quality of collected data, leading to more robust and reliable conclusions about contact patterns. Whilst we aggregate results by income strata due to the limited availability of data (particularly in LICs and middle-income countries), it is important to note that the outcomes considered here are likely to be shaped by several different factors other than country-level income. Whilst some of these factors will be correlated with a country's income status (e.g. household size *Walker et al., 2020*), many others will be unique to a particular setting or geographical area or correlate only weakly with country-level data. Examples include patterns of employment, the role of women, and other contextual factors. These analyses are therefore intended primarily to provide indications of prevailing patterns, rather than a definitive description of contact patterns in a specific context and highlight the significant need for further studies to be carried out in a diversity of different locations.

Despite these limitations however, our results highlight significant differences in the structure and nature of contact patterns across settings. These differences suggest that the comparative importance of different locations and age groups to transmission will likely vary across settings and have critical consequences for the efficacy and suitability of strategies aimed at controlling the spread of respiratory pathogens such as SARS-CoV-2. Most importantly, our study highlights the limited amount of work that has been undertaken to date to better understand and quantify patterns of contact across a range of settings, particularly in LICs and middle-income countries, which is vital in informing control strategies reducing the spread of such pathogens.

## Acknowledgements

We would like to acknowledge the Fiji Ministry of Health and Medical Services for their contribution to the study set in Fiji, M Elizabeth Halloran for sharing the Senegal data, and Nickson Murunga for processing the data request for the Kenyan survey. AM, PW, PGTW, and CW acknowledge joint centre funding from the UK Medical Research Council and DFID (MR/R015600/1). OJW acknowledges funding from the UK Foreign Commonwealth and Development Office. KOK acknowledges support by CUHK Direct grant for research (2019.020), Health and Medical Research Fund (reference number: INF-CUHK-1, 17160302, 18170312), General Research Fund (reference number: 14112818), Early Career Scheme (reference number: 24104920) and Wellcome Trust (United Kingdom, 200861/Z/16/Z). PJD was supported by a fellowship from the UK Medical Research Council (MR/P022081/1); this UK-funded award is part of the European and Developing Countries Clinical Trials Partnership 2 (EDCTP2) programme supported by the EU. EFGN holds an Australian Government Research Training Program Scholarship. FMR receives funding from the Australian National Health and Medical Research Council, WHO, the Bill & Melinda Gates Foundation; Wellcome Trust, DFAT. MM

acknowledges funding from the EPSRC through the EPSRC Centre for Doctoral Training in Modern Statistics and Statistical Machine Learning. JDS received funding for this work from the University of Washington and a grant from US National Institutes of Health, NIAID. CGG declares funding from NIH (K24AI148459). GEP was supported previously by General Medical Sciences/National Institute of Health U01-GM070749. GEP was employed by the Emmes Company while analysing the Niakhar Senegal social contact network data included in this study. The Emmes Company was contracted to perform data cleaning and data analysis of the Niakhar, Senegal clinical trial data (but not the social contact network data) for this study before GEP joined the Emmes Company (in October 2015). After GEP joined the Emmes Company, the sole support from Emmes for this manuscript was in the form of salary support for GEP. The funders had no role in study design, data collection and analysis, decision to publish, or preparation of the manuscript.

## Additional information

### Competing interests

Peter Winskill: received funding from WHO and the Asian Development Bank via their institution to investigate COVID19 Vaccine modelling. The author has no other competing interests to declare. Marco Ajelli: has received research funding from Seqirus outside the submitted work. Carlos G Grijalva: has received funding from NIH, CDC, AHRQ, FDA, Campbell Alliance/Syneos Health and Sanofi, outside the current work. CGG has also received consulting fees from Pfizer, Merck and Sanofi. The author has no other competing interests to declare. Gail E Potter: was employed by the Emmes Company while analyzing the Niakhar Senegal social contact network data included in this study. The Emmes Company was contracted to perform data cleaning and data analysis of the Niakhar, Senegal clinical trial data (but not the social contact network data) for this study before GEP joined the Emmes Company (in October 2015). After GEP joined the Emmes Company, the sole support from Emmes for this manuscript was in the form of salary support for GEP. GEP also received travel support from Thomas Francis Jr. Travel Fellowship. The author has no other competing interests to declare. Jonathan D Sugimoto: received travel support for this work from Thomas Francis, Jr. Scholarship Fund. JDS has also received consultancy fees from the International Vaccine Institute, Seoul, Republic of Korea, for work related to studies of typhoid fever burden and vaccine impact. JDS also participates on a Data Safety Monitoring Board for optimising infant immunisation schedules in Uganda & Nepal, University of Oxford, UK (ongoing). Patrick Walker: received consultancy fees from Pfizer for a lecture on statistical inference using infectious disease models and payment or honoraria from IS Global, Barcelona (for a lecture on statistical inference using infectious disease models) and Lancet Global Health (for reviewing an article). The author has no other competing interests to declare. The other authors declare that no competing interests exist.

### Funding

| Funder | Grant reference number | Author |
|---|---|---|
| Joint Centre funding from the UK Medical Research Council and DFID | MR/R015600/1 | Andria Mousa Peter Winskill Patrick Walker Charles Whittaker |
| Chinese University of Hong Kong | Direct grant for research 2019.020 | Kin O Kwok |
| Health and Medical Research Fund | INF-CUHK-1 | Kin O Kwok |
| General Research Fund of Shanghai Normal University | 14112818 | Kin O Kwok |
| Early Career Scheme | 24104920 | Kin O Kwok |
| Wellcome Trust | 200861/Z/16/Z | Kin O Kwok |

| Funder | Grant reference number | Author |
| --- | --- | --- |
| Medical Research Council | MR/P022081/1 | Peter J Dodd |
| Australian Government Research Training Program Scholarship | | Eleanor FG Neal |
| National Health and Medical Research Council | | Fiona M Russell |
| World Health Organization | | Fiona M Russell |
| Wellcome Trust | | Fiona M Russell |
| Department of Foreign Affairs and Trade | | Fiona M Russell |
| Engineering and Physical Sciences Research Council | | Mélodie Monod |
| University of Washington | | Jonathan D Sugimoto |
| National Institute of Allergy and Infectious Diseases | | Jonathan D Sugimoto |
| National Institutes of Health | K24AI148459 | Carlos G Grijalva |
| National Institutes of Health | U01-GM070749 | Gail E Potter |
| Emmes Company | | Gail E Potter |
| Health and Medical Research Fund | 17160302 | Kin O Kwok |
| Health and Medical Research Fund | 18170312 | Kin O Kwok |

The funders had no role in study design, data collection and interpretation, or the decision to submit the work for publication.

## Author contributions

Andria Mousa, Charles Whittaker, Conceptualization, Formal analysis, Investigation, Methodology, Project administration, Software, Supervision, Validation, Visualization, Writing - original draft, Writing – review and editing; Peter Winskill, Oliver John Watson, Patrick Walker, Conceptualization, Investigation, Methodology, Writing – review and editing; Oliver Ratmann, Mélodie Monod, Investigation, Methodology, Writing – review and editing; Marco Ajelli, Aldiouma Diallo, Peter J Dodd, Carlos G Grijalva, Moses Chapa Kiti, Anand Krishnan, Rakesh Kumar, Supriya Kumar, Kin O Kwok, Claudio F Lanata, Olivier Le Polain de Waroux, Kathy Leung, Wiriya Mahikul, Alessia Melegaro, Carl D Morrow, Joël Mossong, Eleanor FG Neal, D James Nokes, Wirichada Pan-ngum, Gail E Potter, Fiona M Russell, Siddhartha Saha, Jonathan D Sugimoto, Wan In Wei, Robin R Wood, Joseph Wu, Juanjuan Zhang, Data curation, Resources, Writing – review and editing

## Author ORCIDs

Andria Mousa https://orcid.org/0000-0001-9406-7481
Oliver John Watson https://orcid.org/0000-0003-2374-0741
Oliver Ratmann https://orcid.org/0000-0001-8667-4118
Kathy Leung https://orcid.org/0000-0003-4777-388X
Alessia Melegaro https://orcid.org/0000-0003-2221-8898
Charles Whittaker https://orcid.org/0000-0002-5003-2575

## Ethics

Human subjects: All original studies included were approved by an institutional ethics review committee. Ethics approval was not required for the present study.

## Decision letter and Author response

Decision letter https://doi.org/10.7554/eLife.70294.sa1
Author response https://doi.org/10.7554/eLife.70294.sa2

## Additional files

### Supplementary files

- Transparent reporting form
- Supplementary file 1. Search string.
- Supplementary file 2. Preferred Reporting Items for Systematic Reviews and Meta-Analyses (PRISMA)-individual participant data (IPD) checklist of items to include when reporting a systematic review and meta-analysis of IPD.
- Supplementary file 3. Risk of bias table (AXIS critical appraisal tool).
- Supplementary file 4. Study additional Information and data assumptions.
- Supplementary file 5. Data dictionary for participant-level data.
- Supplementary file 6. Extraction table of study characteristics.
- Supplementary file 7. Data availability by study.

### Data availability

All individual-level data across all studies and analysis code are available at https://github.com/mrc-ide/contact_patterns (copy archived at swh:1:rev:0b732099d66b2788ae6da5cf0e8185b25de70868) (see Supplementary file 5 for data dictionary).

The following previously published datasets were used:

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

## Appendix 1

### Systematic review methods and additional information

The search string used to identify eligible studies is shown in *Supplementary file 1*. All search terms were searched in all fields.

The Preferred Reporting Items for Systematic Reviews and Meta-Analyses (PRISMA) checklist of items specific to IPD meta-analyses relevant to this study can be found in *Supplementary file 2*. We assessed the risk of bias using the AXIS critical appraisal tool (*Supplementary file 3*).

Additional information on each study and specific data assumptions are provided in *Supplementary file 4*. The data dictionary associated with the combined dataset shared on github (https://github.com/mrc-ide/contact_patterns) is shown in *Supplementary file 5*.

A1–A3 denote new items that are additional to standard PRISMA items. A4 has been created as a result of re-arranging content of the standard PRISMA statement to suit the way that systematic review IPD meta-analyses are reported.

Reproduced with permission of the PRISMA IPD Group, which encourages sharing and reuse for non-commercial purpose.

### Systematic review findings

A total of 3409 titles and abstracts were retrieved from the databases, and 313 full-text articles were screened for eligibility (*Appendix 1—figure 1*). This search identified 19 studies with suitable contact data from LIC, LMIC, and UMIC settings. Details of the identified studies can be found in *Supplementary file 6*. One study identified in the systematic review and included in the IPD meta-analysis was conducted in both an LMIC (Zambia) and an UMIC (South Africa) setting (*Dodd et al., 2016*). The studies for which authors did not respond (n = 3) included a contact study of infants (*Oguz et al., 2018*). One contact survey with available data was explored qualitatively and was not included in the meta-analysis as the study focused on meal contacts only (*Watson et al., 2017*). For the IPD meta-analysis there were 11 additional datasets from HICs (*Kwok et al., 2018*; *Kwok et al., 2014*; *Leung et al., 2017*; *Mossong et al., 2008*) included which are not shown in the PRISMA diagram (*Appendix 1—figure 1*).

The majority of the studies collected data representative of the general population, some of which were conducted in rural sites (*Horby et al., 2011*; *Kiti et al., 2014*; *le Polain de Waroux et al., 2018a*), urban sites (*Mahikul et al., 2020*; *Zhang et al., 2020*), or a combination of both (*Dodd et al., 2016*; *Melegaro et al., 2017*; *Neal et al., 2020*; *Read et al., 2014*) (rurality was not explicitly stated for all studies). Some of the studies were carried out in the context of another study or trial, such as a flu vaccine trial in Senegal (*Potter et al., 2019*), the community-based nasopharyngeal carriage surveys in Fiji (*Dunne et al., 2018*; *Neal et al., 2020*) and Uganda (*le Polain de Waroux et al., 2018a*), the Study of Respiratory Infections in Andean Peruvian children (RESPIRA PERU; *Grijalva et al., 2015*) and the Manicaland HIV/STD Prevention Study (*Melegaro et al., 2017*).

Overall, the age range of participants was 0–105 years. Although most studies included respondents of all ages, one study restricted their participants to ages over 18 years (*Dodd et al., 2016*), one to ages over 15 years (*Mahikul et al., 2020*), one to ages over 6 months (*Huang et al., 2020*), one study only collected contact data on infants under 6 months (*Oguz et al., 2018*) and another on contacts of children under 6 years and their caregivers (*Neal et al., 2020*). The distribution of participant age groups in each study was also dependent on the sampling method. For instance, two studies focused on school and university students and their contacts, thereby oversampling older children and young adults (*Ajelli and Litvinova, 2017*; *Stein et al., 2014*). In an online survey conducted in Thailand (*Stein et al., 2014*), further participants were invited from convenience samples of university students (snowball sampling) and another study conducted in Russia recruited mostly students and one of their parents (*Ajelli and Litvinova, 2017*). Purposive or quota sampling was also used in four of the studies; one study explored contact patterns of migrant workers in Thailand (*Mahikul et al., 2020*), another of children and their caregivers in Fiji (*Neal et al., 2020*), and two studies, one in Hong Kong (*Leung et al., 2017*) and one in Russia (*Ajelli and Litvinova, 2017*) where children under 18 and university students were oversampled. Most studies (N = 10) deployed a random sampling method (e.g. through population registers), often stratified to include sufficient numbers for each age group. Five studies used a convenience

sample (*Grijalva et al., 2015*; *Kumar et al., 2018*; *Potter et al., 2019*; *Stein et al., 2014*; *Zhang et al., 2020*) and one had no description of sampling methods (*Meeyai et al., 2015*).

Sampling weights to account for selection bias were used by 10 of the studies to account for oversampling or under-sampling particular characteristics. Using inverse probability weights, most studies adjusted for the age and gender structure of the target population using national census data (*Ajelli and Litvinova, 2017*; *Dodd et al., 2016*; *Horby et al., 2011*; *le Polain de Waroux et al., 2018a*; *Leung et al., 2017*; *Melegaro et al., 2017*; *Mossong et al., 2008*; *Potter et al., 2019*; *Stein et al., 2014*). Less often, studies accounted for selection bias in the level of education (*Stein et al., 2014*), rurality (*Kiti et al., 2014*; *Neal et al., 2020*), and household size (*Horby et al., 2011*; *Mossong et al., 2008*; *Potter et al., 2019*; *Stein et al., 2014*). Weights were calculated either in a one-step or two-step approach, depending on the sampling design (e.g. two-stage or stratified design). However, these weights were sometimes not included in the main analysis of a study. A study in Kenya (*Kiti et al., 2014*) accounted for oversampling of semi-urban locations and under-sampling of rural locations, though differences on the estimated contact rates were negligible with the use of weights. In another study which oversampled school- and university-aged students found no substantial effect of accounting for oversampling (*Ajelli and Litvinova, 2017*). Studies using random sampling, such as one conducted in China (*Zhang et al., 2020*), ensured that the age and gender structure of the sample was not significantly different to the one of the general population.

Participant response rates were typically high but variable, ranging from 50% to 98%. All studies reported the number of contacts made in the past 24 hr of (or day preceding) the survey, with some studies reporting the number of contacts over 2 (*Melegaro et al., 2017*) or 3 days (*Potter et al., 2019*). The definitions of contacts were broadly similar (*Supplementary file 6*). Specifically, contacts were defined as skin-to-skin (physical) contact or a two-way conversation in the physical presence of another person, with some studies specifying a minimum duration, distance of contact or number of words exchanged. Ten of the studies identified had a retrospective design without the use of diaries, such as an interview-based questionnaire or online survey. Interviews were conducted both face-to-face and over the phone. Eight studies adopted a diary-based design only, often reporting contacts prospectively, and one study included both interview and diary-based methods (*Zhang et al., 2020*; *Supplementary file 6* ). Data availability by outcome, study-, participant- and contact-level characteristics are shown in *Supplementary file 7*. All studies scored above 65% of the items on the AXIS risk of bias tool, suggesting good or fair quality (*Supplementary file 3*).

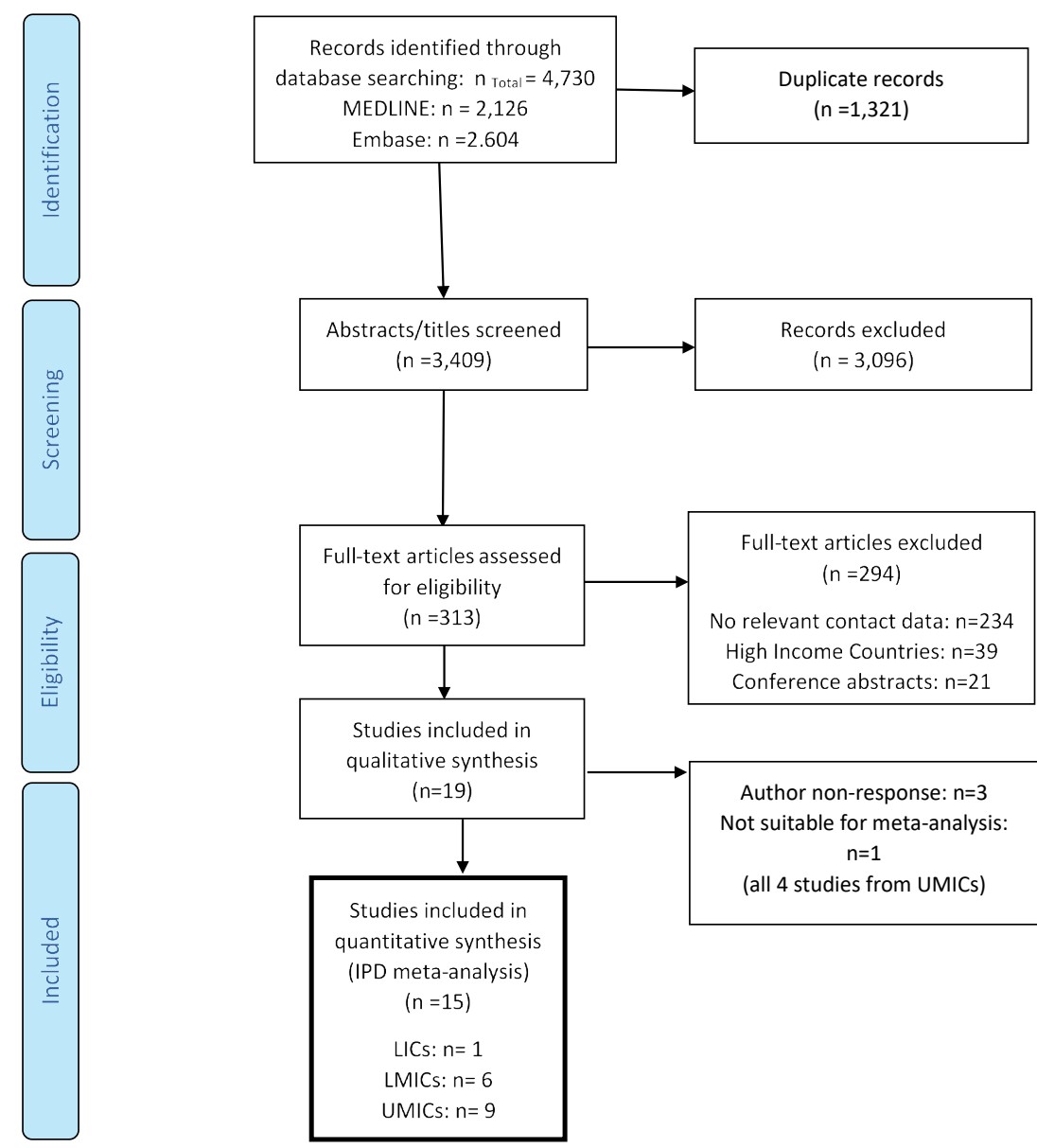

**Appendix 1—figure 1.** Preferred Reporting Items for Systematic Reviews and Meta-Analyses (PRISMA) flow diagram of the screening process and selection of eligible.

## Appendix 2

### Additional IPD results and sensitivity analyses

The median number of total daily contacts by gender, day types, participant age, household size, survey methodology, and student/employment status is shown in *Appendix 2—figure 1*. Boxes indicate the interquartile range. The overlaid violin plots indicate the probability density of the data at different values.

The total number of contacts made at home was proportional to the participant's household size (*Appendix 2—figure 2*). This figure shows the ratio of median number of home contacts to household size, with the shaded area denoting the interquartile range of the ratio. Ratios of >1 (y-axis) indicate more home contacts than household members.

The sensitivity analysis shown in *Appendix 2—figure 3* excluded additional contacts such as additional work contacts, group contacts, and number missed out, which were recorded separately and in less detail by participants compared to their other contacts (*Ajelli and Litvinova, 2017*; *Kumar et al., 2018*; *Leung et al., 2017*; *Zhang et al., 2020*). The coefficients being plotted are those reported in the forest plots in *Figure 1*. In all instances, coefficients were strongly correlated, with Pearson's correlation values of 0.77, 0.97, and 0.99 for LICs/LMICs, UMICs, and HICs, respectively.

The relationships between proportion of contacts made at each location and the different participant or survey characteristics (i.e. age, gender, household size, day of the week, employment/student status) are shown in *Appendix 2—figure 4*. Stacked bar charts for the absolute number of daily contacts by location are shown in *Appendix 2—figure 5*. The relationship between contact location with (a) the proportion of contacts which were physical, (b) the proportion of contacts which lasted a minimum of 1 hr are shown *Appendix 2—figure 6*.

A sensitivity analysis was used to compare all estimated coefficients in the main analysis to an analysis weighting each study equally within an income group (*Appendix 2—figure 7*). This sensitivity analysis uses the same total sample size for an income group and weighs each study equally within an income group, irrespective of its observed sample size. The coefficients being plotted in this figure are those reported in the forest plots in *Figures 1, 3 and 4* in the main text, for total daily contacts (a), whether a contact was physical (b), and duration of contacts (c).

In the weighted analysis, a reverse effect was observed for methodology and the effect of weekday in the total number of daily contacts for UMICs, as a result of disparities in each study's sample size. In the weighted analysis more contacts were observed in interview surveys as compared to diary-based studies (CRR = 1.08, 95% CrI: 1.04–1.13), whereas in the main analysis less contacts were observed in interview surveys (CRR = 0.73, 95% CrI: 0.70–0.77). In UMICs, the number of contacts made on weekdays was higher than those made on weekends in the main analysis (CRR = 1.09, 95% CrI:1.02–1.17), but in the weighted analysis the opposite was true (CRR = 0.92, 95%CrI = 0.87–0.98). For the remaining coefficients, only small quantitative changes (in the effect sizes), but no qualitative changes were observed. In all instances, coefficients were strongly correlated, with Pearson's rho correlation values ranging between 0.93 and 1.00, depending on outcome and income group (see *Appendix 2—table 1*).

**Appendix 2—table 1.** Correlation (Pearson's rho) between coefficients estimated in the main analysis and those from the sensitivity analysis weighing each study equally within an income group.

|  | Income group | Pearson's rho correlation coefficient |
|---|---|---|
|  | LIC/LMIC | 0.927 |
|  | UMIC | 0.962 |
| Total daily contacts | HIC | 0.996 |
|  | LIC/LMIC | 0.987 |
|  | UMIC | 0.984 |
| Duration of contacts | HIC | 0.998 |
|  | LIC/LMIC | 0.977 |
|  | UMIC | 0.974 |
| Physical contacts | HIC | 0.998 |

## Assortativity analysis

Among all studies included in the IPD meta-analysis, 12 studies had collected information on the gender of the contact and 15 studies on the contact's age (*Supplementary file 7*). Eight of those studies provided the contact's exact age or a minimum and maximum estimate, where exact age was unknown. In the remaining seven studies, contact age was provided as predefined categories which varied across studies.

There were three broad contact age categories: 1='children' aged 0 to 12–15, 2='younger adults' aged 13–16 to 40–49 and 3='older adults' ages 41–49 to maximum age. The minimum and maximum for each broad age category is given as a range instead of fixed values to utilise data from all studies providing any information on contact age. Age of contact was usually given as a category, and these categories were different for each study. Participant age groups were the following, where exact age was known: <15, 15 to <45, 45+.

Assortativity was explored in two ways: (a) Each participant in the combined data contributing equally to the matrix proportions (thereby implicitly weighing by study size; *Appendix 2—table 2*) and (b) each study contributing equally to the matrix proportions presented (*Appendix 2—table 3*).

### Method A

Each cell ($m_{r,c}$) in the matrix is defined as the mean proportion of contacts a respondent in age group $r$ makes with a contact in age group $c$. This weighs each respondent equally and does not take study into account.

$i$ indicates the index of the respondent/participant
$r$ indicates the age group of the respondent (1–3)
$c$ indicates the age group of the contact (1–3)

$$m_{r,c} = \frac{1}{n_r} \times \sum_{i=1}^{n_r} p_{r,c,i}$$

where
$n_r$ = total number of participants in age group $r$
$p_{r,c,i}$ = proportion of contacts that are in age group $c$ among all contacts made by the $i$th participant in age group $r$

**Appendix 2—table 2.** Assortativity by age and sex, weighing by study sample size (method A).

| | | Age category | | | | | Gender | |
|---|---|---|---|---|---|---|---|---|
| **LIC/LMIC** | | | | | | | | |
| | | Contact age | | | | | Contact gender | |
| | | 1 | 2 | 3 | | | Male | Female |
| | 1 | 0.47 | 0.41 | 0.12 | | Male | 0.59 | 0.41 |
| | 2 | 0.22 | 0.64 | 0.14 | Participant gender | Female | 0.41 | 0.59 |
| Participant age | 3 | 0.20 | 0.51 | 0.29 | | | | |
| **UMIC** | | | | | | | | |
| | | Contact age | | | | | Contact gender | |
| | | 1 | 2 | 3 | | | Male | Female |
| | 1 | 0.34 | 0.51 | 0.15 | | Male | 0.52 | 0.48 |
| | 2 | 0.20 | 0.62 | 0.17 | Participant gender | Female | 0.46 | 0.54 |
| Participant age | 3 | 0.14 | 0.41 | 0.45 | | | | |
| **HIC** | | | | | | | | |
| | | Contact age | | | | | Contact gender | |
| | | 1 | 2 | 3 | | | Male | Female |
| | 1 | 0.55 | 0.31 | 0.14 | | Male | 0.51 | 0.49 |
| | 2 | 0.24 | 0.53 | 0.23 | Participant gender | Female | 0.42 | 0.58 |
| Participant age | 3 | 0.15 | 0.33 | 0.51 | | | | |

## Method B

Each cell ($M_{r,c}$) in the matrix is defined as the mean proportion a respondent in age group $r$ makes with a contact in age group $c$.

$s$ indicates index of the study

$$M_{r,c} = \frac{1}{N} \times \sum_{s=1}^{N} \left[ \frac{1}{n_{r,s}} \times \sum_{i=1}^{n_{rs}} p_{r,c,i,s} \right]$$

where

$n_{r,s}$ = total number of participants in age group $r$ in study $s$

$N$ = total number of studies

$p_{r,c,i,s}$ = proportion of contacts that are in age group $c$ among all contacts made by the $i$th participant in age group in study $s$

**Appendix 2—table 3.** Assortativity by age and sex, weighing each study equally (method B).

| | Contact age | | | | Contact gender | |
|---|---|---|---|---|---|---|
| **LIC/LMIC** | | | | | | |
| | 1 | 2 | 3 | | Male | Female |

*Appendix 2—table 3 Continued on next page*

*Appendix 2—table 3 Continued*

|  |  | Contact age | | | |  | Contact gender | |
|---|---|---|---|---|---|---|---|---|
|  | 1 | 0.48 | 0.41 | 0.11 |  | Male | 0.55 | 0.45 |
|  | 2 | 0.27 | 0.60 | 0.13 | Participant gender | Female | 0.42 | 0.58 |
| Participant age | 3 | 0.21 | 0.51 | 0.28 |  |  |  |  |

### UMIC

|  |  | Contact age | | |  | Contact gender | |
|---|---|---|---|---|---|---|---|
|  |  | 1 | 2 | 3 |  | Male | Female |
|  | 1 | 0.38 | 0.46 | 0.16 |  | Male | 0.54 | 0.46 |
|  | 2 | 0.21 | 0.61 | 0.18 | Participant gender | Female | 0.44 | 0.56 |
| Participant age | 3 | 0.18 | 0.50 | 0.32 |  |  |  |  |

### HIC

|  |  | Contact age | | |  | Contact gender | |
|---|---|---|---|---|---|---|---|
|  |  | 1 | 2 | 3 |  | Male | Female |
|  | 1 | 0.54 | 0.31 | 0.14 |  | Male | 0.51 | 0.49 |
|  | 2 | 0.27 | 0.51 | 0.22 | Participant gender | Female | 0.42 | 0.58 |
| Participant age | 3 | 0.21 | 0.31 | 0.48 |  |  |  |  |

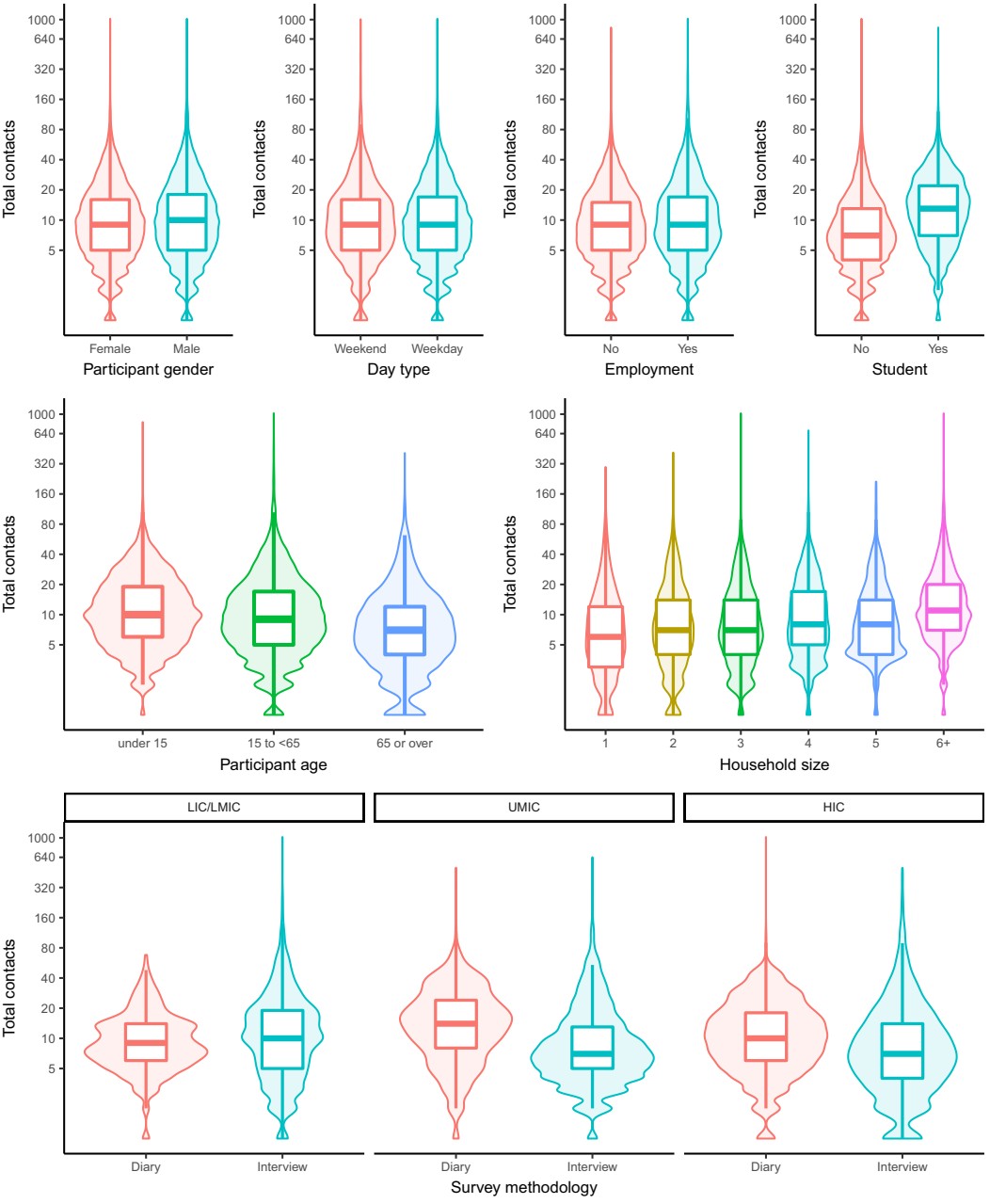

**Appendix 2—figure 1.** Total number of contacts boxplots and violin plots by participant/study characteristics.

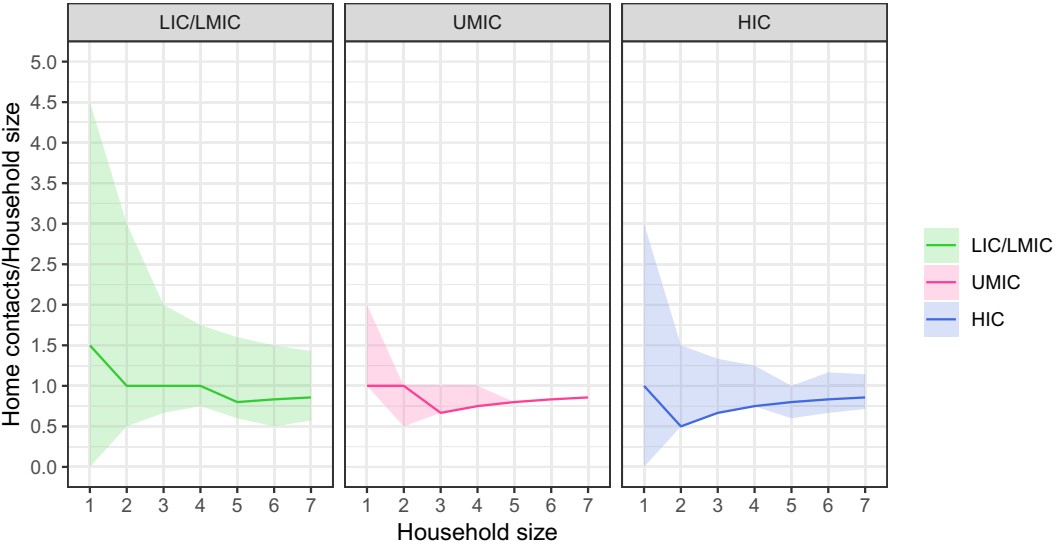

**Appendix 2—figure 2.** The relationship between household size and median daily contacts made at home divided by a participant's household size, stratified by income strata.

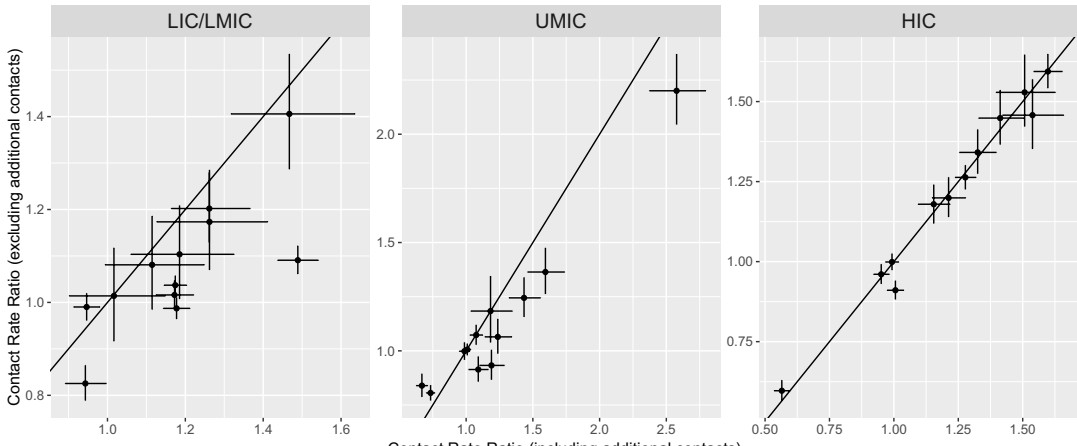

**Appendix 2—figure 3.** Comparison of estimated regression coefficients for predicting total contacts with and without the inclusion of additional contacts.

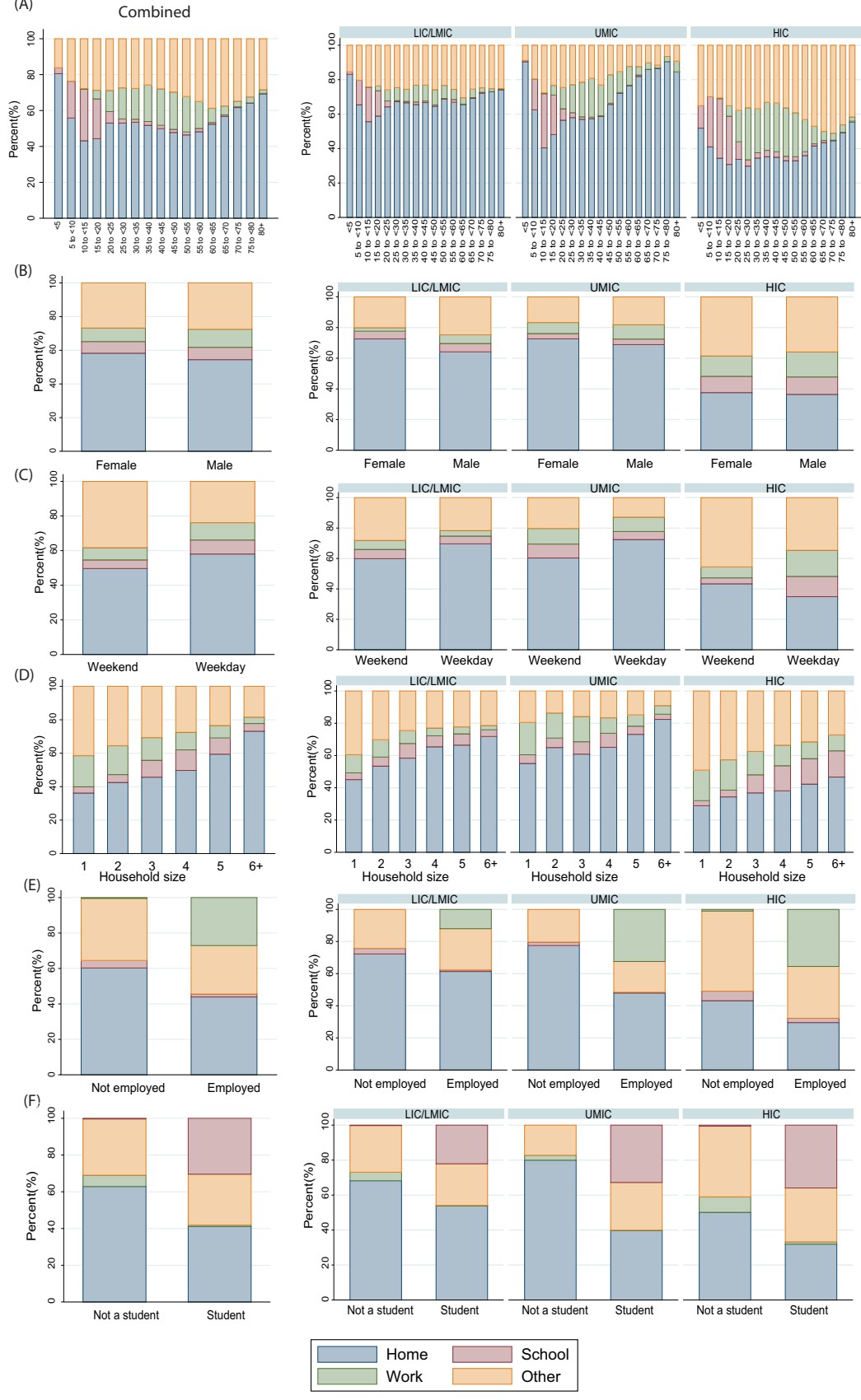

**Appendix 2—figure 4.** Location of contacts as a percentage of total daily contacts by (**A**)

*Appendix 2—figure 4 continued on next page*

*Appendix 2—figure 4 continued*

participant's age, (**B**) participant's gender, (**C**) day of the week, (**D**) household size, (**E**) employment status (in participants aged 18 or over), and (**F**) student status in participants aged 5 to <20 years.

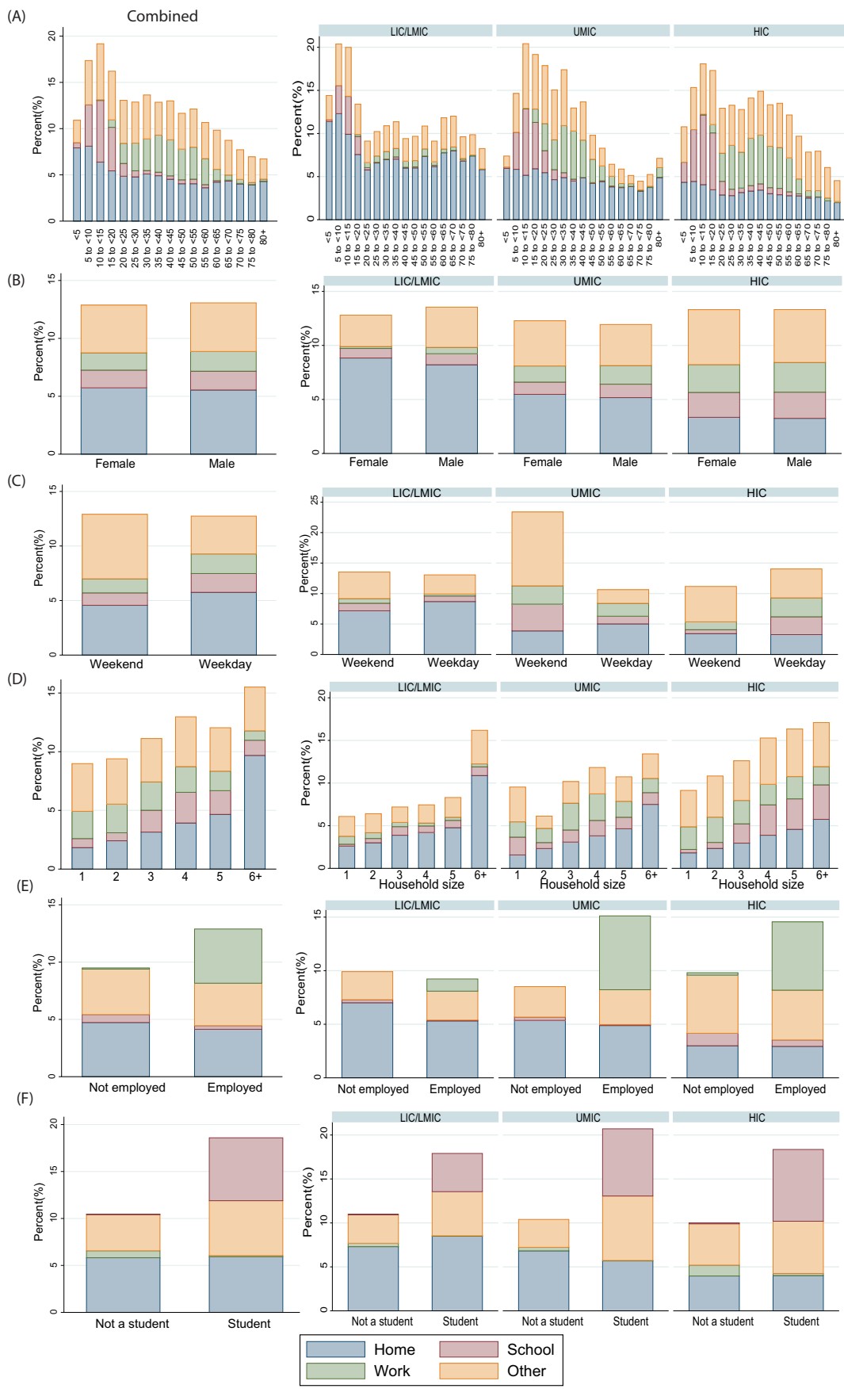

**Appendix 2—figure 5.** Total number of daily contacts in each location by (**A**) participant's age, (**B**) participant's gender, (**C**) day of the week, (**D**) household size, (**E**) employment status (in participants aged 18 or over), and (**F**) student status in participants aged 5 to < 20 years.

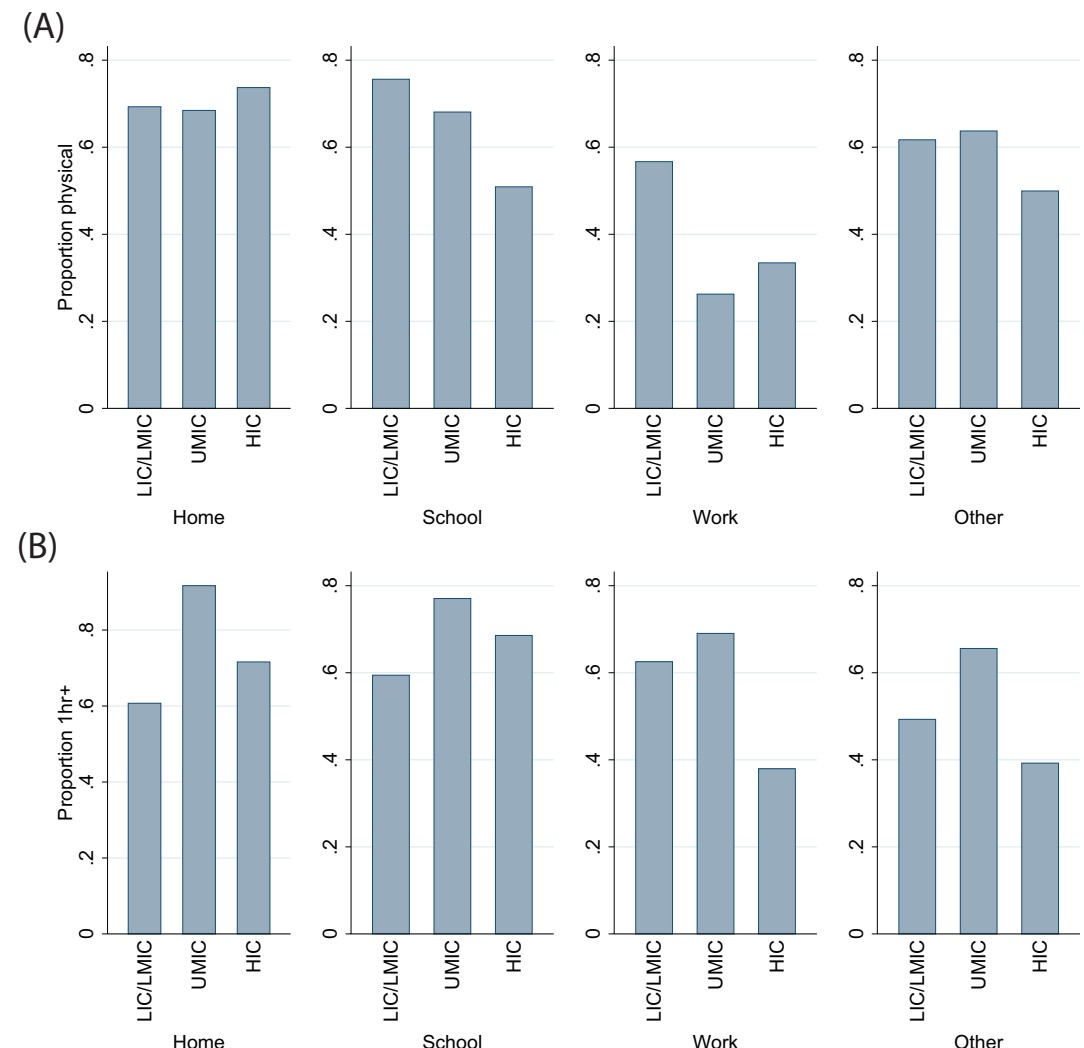

**Appendix 2—figure 6.** Contact location and (A) type of contacts and (B) duration of contact, by income group.

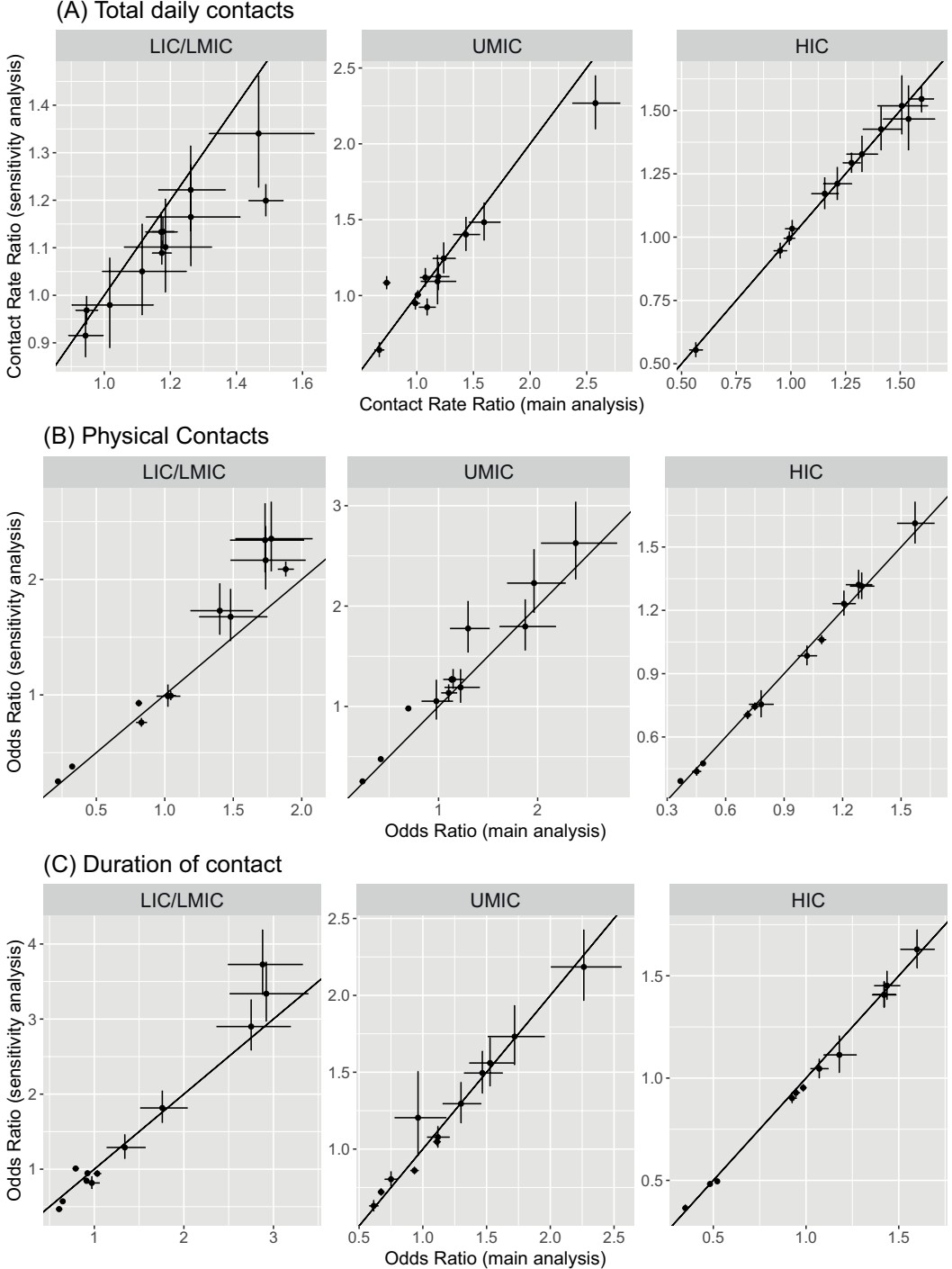

**Appendix 2—figure 7.** Comparison of estimated regression coefficients in the main analysis and sensitivity analysis weighing each study equally within an income group.

