## [Decision Letter]

**Decision letter after peer review:**

Thank you for submitting your article "Social Contact Patterns and Implications for Infectious Disease Transmission: A Systematic Review and Meta-Analysis of Contact Surveys" for consideration by *eLife*. Your article has been reviewed by 2 peer reviewers, and the evaluation has been overseen by a Reviewing Editor and David Serwadda as the Senior Editor. The reviewers have opted to remain anonymous.

Essential revisions:

1) Despite the title of the paper "Social Contact Patterns and Implications for Infectious Disease Transmission: A Systematic Review and Meta-Analysis of Contact Surveys", the focus is on the contact patterns, and the implications for infectious disease transmission are minimally discussed. The paper needs more discussion on the implications of the findings.

2) The authors mentioned that all analyses were stratified by income strata. Therefore, it is my understanding that they pooled the individual level data from different studies, with very different sample sizes together (see Table 1). My concern is that the authors do not mention use of "weights" in the paper or in the supplement. Did the datasets include sampling weights? And if the datasets do contain weights, were the original weights used in the analyses or did the authors reweigh the data? If the authors don't use any weights how to they account for the fact that some countries had much larger sample sizes which might influence the results?

*Reviewer #1:*

The strength of this manuscript is the comparison between different income settings and the focus on the heterogeneity in the number of contacts in total, by age and for example by location. The meta-analysis is conducted in a systematic way and results are provided for may sub-categories. This work highlights the need to incorporate setting-specific contact patterns when using contact patterns for disease modelling to e.g. inform policy making.

The pitfall of this manuscript is the uniform analysis of raw data from different social contact studies, which all have differences in study design. E.g. The study from Dodd et al., in Zambia reports a median number of 4 daily contacts, though they only surveyed adults. Other surveys are on the population level or might have included only school children? The distinction by survey type (diary based / interview) is very informative, though more information on the survey design and study population in the main text might be useful for the reader.

The differences in social contact patterns are extensively described, though the discussion of the implications regarding transmission dynamics is minimal. At least, I expected more given the explicit referral in the title to “… implications for infectious disease transmission”. What do the differences that have been observed mean in terms of (air-born) disease transmission? Would you recommend different NPI’s for different countries based on this study? Previous studies found that social contact data are a good proxy for disease transmission. Though, the proportionality factor to link social contacts to transmission events, can be setting specific, so that a different number of contacts might lead to the same number of transmission events. Other lessons learned going from the heterogeneous contact patterns to transmission dynamics?

The number of household members is positively correlated with the number of contacts at home, though is this also the case if you analyse this per household member? As such, have individuals in large households more visitors at home than e.g. people in a two-person household? This could be relevant for transmission dynamics since having contact every day with the same household members, irrespectively of the number, is not a large risk to contract a disease. On the other hand, having many different contacts, in so-called open contact networks, is more in favour of infectious disease transmission. In-dept network analyses are not possible with the data at hand, and outside the scope of this work, though more discussion on the link between transmission and social contact would be an added value for this work.*Reviewer #2:*

Summary:

The authors obtained social contact data from 19 studies (27 individual surveys) that included countries from different income strata and compared the social contact and mixing patterns. The review includes data from one low-income country (LIC), six lower-middle income countries (LMIC), and nine studies from upper-middle income countries (UMIC) and four studies (11 surveys) high income settings all prior to the COVID-19 pandemic.

These studies were included because data was available and they used similar methodology, and included information on total number of contacts, contacts age and gender. Many studies also contained other variables of interests such as type and duration of contact. However, there were some differences; for instance, diary-based surveys versus interview/questionnaire-based surveys.

The authors find that there are systematic differences, especially in the age pattern and location of contacts by countries’ income strata. Gender and type of day (weekday vs weekend) also has differential effect on contact patterns based on countries’ income strata. These differences imply that the effectiveness of different non-pharmaceutical interventions will vary across countries.

Strengths:

The authors obtained and analyzed participants social contact data from 27 surveys. Their dataset contained information on 28,000 individuals and over 400,000 contacts. It’s a huge feat to standardize the datasets especially when some studies captured information on group contacts differently. The authors used appropriate model to study association between study/participant characteristics, and total number, duration, and type of daily contacts.

Weaknesses:

The authors mentioned that all analyses were stratified by income strata. Therefore, it is my understanding that they pooled the individual level data from different studies, with very different sample sizes together (see Table 1). My concern is that the authors do not mention use of “weights” in the paper or in the supplement. Did the datasets include sampling weights? And if the datasets do contain weights, were the original weights used in the analyses or did the authors reweigh the data? If the authors don’t use any weights how to they account for the fact that some countries had much larger sample sizes which might influence the results?

The social contact data were all collected prior to the COVID-19 pandemic between 2005-2018. Should we be concerned that during the 13-year span there may be some time trends in contact patterns?

Can you provide some justification for choosing 1 hour for the cutoff in the duration analyses?

What do you mean by included random study effects (line 359)?

In line 365, you mention that you report CRR and Odds Ratios that are adjusted for gender and age. I take it to mean that you ran separate regressions for each of the key variables (age group, gender, weekday, …) and controlled for age and/or gender depend on variable. Is that correct? If that is the case, are the rate ratios in figures 1,3,4 all adjusted? If that is the case, can you mention it in the figure caption.

---

## [Author Response]

Reviewer #1:The strength of this manuscript is the comparison between different income settings and the focus on the heterogeneity in the number of contacts in total, by age and for example by location. The meta-analysis is conducted in a systematic way and results are provided for may sub-categories. This work highlights the need to incorporate setting-specific contact patterns when using contact patterns for disease modelling to e.g. inform policy making.The pitfall of this manuscript is the uniform analysis of raw data from different social contact studies, which all have differences in study design. E.g. The study from Dodd et al., in Zambia reports a median number of 4 daily contacts, though they only surveyed adults. Other surveys are on the population level or might have included only school children? The distinction by survey type (diary based / interview) is very informative, though more information on the survey design and study population in the main text might be useful for the reader.

We thank the reviewer for their comments. Indeed, there were differences in study design and populations which were included. Some of the systematic review findings, and particularly contextual differences, as well as study design (eg. sampling/methodology) and populations included by the different studies were originally detailed in Supplementary Text 1 (in the first submission – now found in Appendix 1) to preserve space in the main text. We have now moved some of this information back into the main text:

“The majority of the studies collected data representative of the general population, through random sampling and included a combination of both rural and urban sites (see Appendix 1 for further details). Although most studies included respondents of all ages, one study restricted their participants to ages over 18 years (Dodd et al., 2015), one to ages over 15 years (Mahikul et al., 2020), one to ages over 6 months (Huang et al., 2020), one study only collected contact data on infants under 6 months (Oguz et al., 2018) and another on contacts of children under 6 years and their caregivers (Neal et al., 2020). The distribution of participant age groups in each study was also dependent on the sampling method. For instance, two studies focused on school and university students and their contacts, thereby oversampling older children and young adults (Ajelli and Litvinova, 2017; Stein et al., 2014).”

The benefit of using an individual- participant data meta-analysis is that we were able to account for factors such as age and study design, in which studies might differ. Other types of unmeasured heterogeneity between studies, eg. contextual factors and study design differences are not fully accounted for, though to some extent would be captured by the study random effects. These limitations are discussed in the second to last paragraph of the discussion.

The differences in social contact patterns are extensively described, though the discussion of the implications regarding transmission dynamics is minimal. At least, I expected more given the explicit referral in the title to "… implications for infectious disease transmission". What do the differences that have been observed mean in terms of (air-born) disease transmission? Would you recommend different NPI's for different countries based on this study? Previous studies found that social contact data are a good proxy for disease transmission. Though, the proportionality factor to link social contacts to transmission events, can be setting specific, so that a different number of contacts might lead to the same number of transmission events. Other lessons learned going from the heterogeneous contact patterns to transmission dynamics?

We thank the reviewer for this important point and agree that more elaboration on possible implications for transmission is required. To that end, we have added the following text to the discussion which attempts to describe some of the possible implications (whilst also couching our conclusions in tentative terms, given the significant limitations the reviewer highlights both above and in their comment below):

On how different household structures might modulate the viability of “shielding” strategies, with additional text added and now citing supporting work examining how different household structures might necessitate different shielding strategies in low income and conflict-affected settings:

“These results have important consequences for the viability and efficacy of protective policies centred around shielding of elderly individuals (i.e. those most at risk from COVID-19 or influenza). In these settings other strategies may be required to effectively shield vulnerable populations, as has been previously suggested (Dahab et al., 2020).”

On how differences in the comparative importance of different locations (household, school etc) might modulate the impact of different NPIs, and how this may limit the generalisability of estimates of NPI effectiveness to date, which has disproportionately been based on European data:

“Altogether, these results suggest differences between settings in the comparative importance of different locations (such as the household or the workplace) to transmission of SARS-CoV-2, a finding which would likely modulate the impact of different NPIs (such as workplace or school closures, stay at home orders etc). Moreover, it suggests that previous estimates of NPI effectiveness primarily derived from European data and settings (Brauner et al., 2020) may be of limited generalisability to non-European settings characterised by different structures and patterns of social contact.”

The number of household members is positively correlated with the number of contacts at home, though is this also the case if you analyse this per household member? As such, have individuals in large households more visitors at home than e.g. people in a two-person household? This could be relevant for transmission dynamics since having contact every day with the same household members, irrespectively of the number, is not a large risk to contract a disease. On the other hand, having many different contacts, in so-called open contact networks, is more in favour of infectious disease transmission. In-dept network analyses are not possible with the data at hand, and outside the scope of this work, though more discussion on the link between transmission and social contact would be an added value for this work.

We thank the reviewer for this suggestion. To address this point we have added an additional supplementary figure (Appendix 2 – Figure 2), where the x-axis is the household size and the y axis is the ratio of the median number of contacts made at home divided by the household size. This plot shows that this median ratio is close to 1 for all household sizes (and across all income groups), highlighting that the main driver of the number of contacts made at home is household size. Although the data cannot distinguish between the number of home contacts who are visitors and those who are household members (and hence this question cannot be answered directly), this sensitivity analysis indicates that it is unlikely that bigger households have more visitors at home compared to smaller households. This can be found in the Results section in the 3^rd^ paragraph under the section “Total number of contacts and contact location” and it is also mentioned in the 3^rd^ paragraph of the discussion.

We completely agree with the reviewer that a limitation of this analysis is our inability to recapitulate the complete social network on which these contacts were being made. We have therefore added in the following additional text into the Discussion highlighting this limitation (and the promise such data could hold):

“However, beyond highlighting heterogeneity in where and how transmission is likely to occur, it remains challenging to disentangle exactly how these differences in contact patterns would shape patterns of transmission. Whilst the collated data provide a cross-sectional snapshot into the networks of social contact underpinning transmission, they remain insufficient to completely resolve the underlying network or its temporal dynamics. Our results therefore do not consider key features relevant to population-level spread and transmission (such as overall network structure or the extent of repeated contacts, which would be most likely to occur with household members) which previous work has demonstrated can have a significant impact on infectious disease dynamics, both in general terms (Keeling and Eames 2005, Bansal et al., 2010) as well as with COVID-19 (Rader et al., 2021). It is in this context that recent results generating complete social networks (including both the frequency and identity of an individual’s contacts) from high-resolution GPS data represent promising developments in understanding social contact networks and how they shape transmission (Firth et al., 2020).”

Reviewer #2:Summary:The authors obtained social contact data from 19 studies (27 individual surveys) that included countries from different income strata and compared the social contact and mixing patterns. The review includes data from one low-income country (LIC), six lower-middle income countries (LMIC), and nine studies from upper-middle income countries (UMIC) and four studies (11 surveys) high income settings all prior to the COVID-19 pandemic.These studies were included because data was available and they used similar methodology, and included information on total number of contacts, contacts age and gender. Many studies also contained other variables of interests such as type and duration of contact. However, there were some differences; for instance, diary-based surveys versus interview/questionnaire-based surveys.The authors find that there are systematic differences, especially in the age pattern and location of contacts by countries' income strata. Gender and type of day (weekday vs weekend) also has differential effect on contact patterns based on countries' income strata. These differences imply that the effectiveness of different non-pharmaceutical interventions will vary across countries.Strengths:The authors obtained and analyzed participants social contact data from 27 surveys. Their dataset contained information on 28,000 individuals and over 400,000 contacts. It's a huge feat to standardize the datasets especially when some studies captured information on group contacts differently. The authors used appropriate model to study association between study/participant characteristics, and total number, duration, and type of daily contacts.Weaknesses:The authors mentioned that all analyses were stratified by income strata. Therefore, it is my understanding that they pooled the individual level data from different studies, with very different sample sizes together (see Table 1). My concern is that the authors do not mention use of "weights" in the paper or in the supplement. Did the datasets include sampling weights? And if the datasets do contain weights, were the original weights used in the analyses or did the authors reweigh the data? If the authors don't use any weights how to they account for the fact that some countries had much larger sample sizes which might influence the results?

We thank the reviewer for their comments and feedback. Our analysis was an individual participant meta-analysis, meaning that each study’s sample size is implicitly weighted by the number of individuals included (ie number of rows).

In total, 10 studies used sampling weights in their original analysis to account for over-sampling or under-sampling particular characteristics. Using inverse probability weights, most studies adjusted for the age and gender structure of the target population using national census data. Less often, studies accounted for selection bias in the level of education, rurality and household size. Weights were calculated either in a one-step or two-stepped approach, depending on the sampling design (eg. two-stage or stratified designs). However, these weights were sometimes not included in the main analysis of a study, and only 1 publication included the sampling weights in the shared dataset. In two studies, where contact rates were compared with and without sampling weights, there was little or no difference. Other studies have also compared the age and gender structure of the target population to that of the study sample, and found no significant difference. These points are now discussed in Appendix 1 in the 4^th^ paragraph under “Systematic review findings”. We expect that differences in sampling bias between studies to be partly accounted for by using random effects.

In response to the last question in the above comment, we have conducted a sensitivity analysis, modelling the main three outcomes (total number of contacts, proportion of contacts lasting >1 hour and proportion of physical contacts), to weigh each included study equally in the analysis (see 2^nd^ paragraph under “statistical analysis” in the Methods section). The findings from this analysis can be found in Appendix 2 – Figure 2 and Appendix 2 – Table 1. This sensitivity analysis yielded similar results to the main analysis, and any differences are discussed in the new supplementary text. Equal weighting by study has also been used in the assortativity analysis by age and gender (Appendix 2 and Appendix 2 – Table 3, method B), and no substantial differences were observed when compared to weighting by study sample size (Appendix 2 and Appendix 2 – Table 2, method A).

The above comment also addresses essential revision #2.

The social contact data were all collected prior to the COVID-19 pandemic between 2005-2018. Should we be concerned that during the 13-year span there may be some time trends in contact patterns?

We thank the reviewer for raising this point – due to the cross-sectional nature of the surveys (and the limited number of surveys collated), we were unable to explore time-trends in detail. We have added the following text to the Discussion highlighting this:

“Relatedly, it is also important to note that the studies collated here were conducted over a wide time-period (2005-2018). In conjunction with the cross-sectional nature of the included studies, this precludes us from being able to examine for potential time-related trends in contact patterns.”

With regards to the dynamic and potentially time-varying nature of contact patterns, we have also added additional citations and text to the section that highlights that contact patterns are likely to have changed in the period since the pandemic commenced.:

“Previous work has documented significant alterations to patterns of social contact in response to individual-level behaviour changes or government implemented NPIs aimed at controlling SARS-CoV-2 spread, and that these changes are dynamic and time-varying (Gimma et al., 2021, McCreesh et al., 2021). A detailed understanding of the impact of changing contact patterns on disease spread necessarily requires both an understanding of baseline contact patterns (as detailed in the studies collated here), and what changes have occurred as a result of control measures – this latter data remains sparse and is available for only a limited number of settings (Jarvis et al., 2021, 2020; Liu et al., 2021).”

Can you provide some justification for choosing 1 hour for the cutoff in the duration analyses?

The reason for using 1 hour as the cut-off point for the analysis on duration is to preserve sample size by allowing inclusion of all available data (each study used different duration categories). This is now mentioned in the methods section, in the second paragraph under “Statistical analysis”.

What do you mean by included random study effects (line 359)?

What we meant by this, is that for all models, we fitted random effects to account for heterogeneity between the included studies. As suggested, we have now edited this sentence (in the last paragraph of the methods section) to improve clarity.

In line 365, you mention that you report CRR and Odds Ratios that are adjusted for gender and age. I take it to mean that you ran separate regressions for each of the key variables (age group, gender, weekday, …) and controlled for age and/or gender depend on variable. Is that correct? If that is the case, are the rate ratios in figures 1,3,4 all adjusted? If that is the case, can you mention it in the figure caption.

We thank the reviewer for this observation. That is correct, all regressions were adjusted for age/ gender and ran separately for each key variable. As suggested, we have now edited all figure legends (in figures 1,3,4) to clarify this by adding the following:

“All models were adjusted for age and gender and were ran separately for each key variable (weekday/weekend, household size, survey methodology, student/employment status).”